# Are seamounts refuge areas for fauna from polymetallic nodule fields?

Daphne Cuvelier1*, Pedro A. Ribeiro1,2*, Sofia P. Ramalho1,3*, Daniel Kersken4,5, Pedro Martinez Arbizu5, Ana Colaço1

MARE – Marine and environmental sciences centre/IMAR – Instituto do Mar/Centro OKEANOS – Universidade dos Açores, Rua Prof. Dr. Frederico Machado 4, 9901-862 Horta, Portugal
Current address: Department of Biological Sciences and K.G. Jebsen Centre for Deep-Sea Research, University of Bergen, Bergen, Norway.
Current address: Departmento de Biologia & CESAM, Universidade de Aveiro, Campus Universitário de Santiago, 3810-193 Aveiro, Portugal
Department of Marine Zoology, Senckenberg Research Institute and Natural History Museum, Senckenberganlage 25, 60325 Frankfurt am Main, Germany
German Centre for Marine Biodiversity Research (DZMB), Senckenberg am Meer, Südstrand 44, 26382 Wilhelmshaven, Germany

* Contributed equally to this work/Corresponding authors: Daphne Cuvelier (daphne.cuvelier@gmail.com), Pedro Ribeiro (Pedro.Ribeiro@uib.no) and Sofia Pinto Ramalho (sofia.pinto.ramalho@gmail.com)

Running title: Seamounts as refuge areas for nodule fauna

Six keywords: megafauna, seamounts, nodule fields, image analysis, deep sea, mining

## Abstract

Seamounts are abundant and prominent features on the deep-sea floor and intersperse with the nodule fields of the Clarion-Clipperton Fracture Zone (CCZ). There is a particular interest in characterising the fauna inhabiting seamounts in the CCZ because they are the only other ecosystem in the region to provide hard substrata besides the abundant nodules on the soft sediment abyssal plains. It has been hypothesised that seamounts could provide refuge for organisms during deep-sea mining actions or that they could play a role in the (re-)colonisation of the disturbed nodule fields. This hypothesis is tested by analysing video transects in both ecosystems, assessing megafauna composition and abundance.

Nine video transects (ROV dives) from two different license areas and one Area of Particular Environmental Interest in the eastern CCZ were analysed. Four of these transects were carried out as exploratory dives on four different seamounts in order to gain first insights in megafauna composition. The five other dives were carried out in the neighbouring nodule fields in the same areas. Variation in community composition observed among and along the video transects was high, with little morphospecies overlap on intra-ecosystem transects. Despite the observation of considerable faunal variations within each ecosystem, differences between seamounts and nodule fields prevailed, showing significantly different species associations characterising them, thus questioning their use as a possible refuge area.

# 1. Introduction

Seamounts are abundant and prominent features on the deep-sea floor (Wessel et al., 2010). They are common in all the world's oceans, occurring in higher abundances around mid-ocean ridges, island-arc convergent areas, and above upwelling mantle plumes (Kitchingman et al., 2007). Seamounts are defined as isolated sub-surface topographic feature, usually of volcanic origin, of significant height above the seafloor (International Seabed Authority (ISA), 2019). They are generally isolated, typically cone shaped undersea mountains rising relatively steeply at least several hundred meters from the deep-sea floor. Seamounts comprise a unique deep-sea environment, characterised by substantially enhanced currents and a fauna that is dominated by suspension feeders, such as corals (Rogers, 2018). They represent hard substrata in the otherwise soft sediment deep sea and can thus be considered habitat islands (Beaulieu, 2001). Given the growing evidence that seamounts differ substantially across a range of spatial scales, the concept of seamounts as a single, relatively well-defined habitat type is outdated (Clark et al., 2012). Depth and substrate type are key elements in determining the composition and distribution of benthic fauna on seamounts, while location is likely the subsequent most important driver of faunal composition and distribution patterns (e.g. Tittensor et al., 2009). Connectivity varies substantially between seamounts, resulting in the presence of taxa with very localised to very wide distributions (Clark et al., 2010).

The Clarion-Clipperton Fracture Zone (CCZ), in the equatorial eastern Pacific Ocean, is most known for its extensive polymetallic nodule fields that will potentially be mined in the future. In this area, nodules represent the most common hard substrate on the soft-sediment abyssal plains, and many organisms rely on them for survival (Vanreusel et al., 2016). Removal of hard substrate through mining actions will impact all these organisms, which were estimated at about 50% of all megafaunal species in the CCZ (Amon et al., 2016). Nodule fields in the CCZ are interspersed by seamounts (Wedding et al., 2013), the only feature offering hard substrata besides the nodules. Based on this feature/characteristic, it has been hypothesised that seamounts could provide refuge for organisms during deep-sea mining activities or that seamounts could play a role in the (re-)colonisation of the disturbed nodule fields. Whether or not this is true may have important implications for management of the impacts of polymetallic nodule mining in the CCZ. However, knowledge on the biodiversity inhabiting seamounts in this region is currently lacking.

The objectives of the current study were twofold: (i) Provide first insights in seamount megafauna within the CCZ, (ii) Compare the benthic fauna inhabiting seamounts and nodule fields in the eastern CCZ. Since this is the first time the seamounts at the eastern CCZ were visited, a separate section is dedicated to describe these first insights.

# 2. Material and Methods

## 2.1. Study site and data

During the SO239 ECORESPONSE cruise in 2015 (Martinez Arbizu and Haeckel, 2015), four seamounts were visited for the first time within two different license areas and one area of particular environmental interest (APEI) within the Clarion-Clipperton Fracture zone (CCZ) (Table 1). Nodule fields within the same license areas were visited and sampled as well. Video imagery and faunal samples were collected by a Remotely Operated Vehicle (ROV Kiel 6000 (GEOMAR), equipped with a high definition Kongsberg OE14-500 camera).

Seamount transects were carried out uphill, towards the summit resulting in a depth gradient along
the transect (Table 1). The four seamount transects were characterised by different depth ranges
and lengths and were, due to the vessel's positioning and the predominant South-East surface
currents, all carried out downstream, on the north to north-western flanks of the seamounts (Table
1 and Fig. 1). The names of the seamounts used here, Rüppel and Senckenberg (BGR, German
License area), Heip (GSR, Belgian License area) and Mann Borgese (APEI3), are the ones agreed upon
by the scientist during the ECORESPONSE cruise (Martinez Arbizu and Haeckel, 2015), pending
incorporation of these names in the GEBCO gazetteer. The seamounts differed in shape and size
with Senckenberg and Heip being a sea-mountain range, while Rüppel and Mann Borgese were more
isolated, stand-alone seamounts (Fig. 1). Nodule field dives were carried out on relatively flat
surfaces (maximum depth range covered during a dive or transect was 30m difference, Table 1) and
were referred to by the dive number and license area. The five nodule transects were all located
between 4000-5000m depth and the transects differed in length between dives as well (Table 1).
Within the same license area, distance between different transects was 16 to 60km, while distance
between license areas added up to several hundreds of kilometres (minimum ~700kms BGR – GSR,
Fig. 1).
Investigated areas were restricted to the eastern part of the CCZ with APEI3 being the most north-
and westward bound area. The optical resolution of the camera enabled reliable identification of
organisms larger than 3 cm (Martinez Arbizu and Haeckel, 2015). The combination of exploration
and opportunistic sampling restricted a systematic image collection. Target ROV travelling altitude
was <2m and travelling speed was~0.2m/s which, along with the camera zoom, were kept constant
whenever possible. Due to the explorative nature of the dives, the pan and tilt of the ROV camera
were not kept constant.
## 2.2. Video analysis and statistics
All videos were annotated to the lowest taxonomic level possible. The number of morphospecies,
defined as morphologically different organisms within the lowest taxonomic group identified, were
assessed. Identifications were double checked with scientists working in the same area as well as
taxonomic experts and comprise different taxonomic levels (e.g. Genus, Family). Those
identifications restricted to higher taxon groups (Family, Class, etc.) and for which it was impossible
to attribute a morphospecies, were referred to as taxa and are likely to morphologically differ
between transects. Xenophyophores, living on the soft sediment deep-sea floor, were less
prominently present at seamounts than at nodule fields and were not quantified. Fish
(Actinopterygii), Crustacea (Nematocarcinidae, Aristeidae, Peracarida) and Polychaeta were
quantified but left out of the comparing statistical analysis due to their lack of representativity and
possible attraction due to ROV lights. The same was done for jellyfish and other doubtful
identifications that could not be confidently assigned to a higher taxonomic group (Table A1). A
subset of the nodule field transects form BGR, GSR and APEI3 was presented by Vanreusel et al.
(2016), corresponding to 44% of what was studied here and limited organism identification to a
higher taxonomic level (Order (e.g. Alcyonacea) or Class (e.g. Ophiuroidea)). In our study, the entire
transects (100%) were annotated to morphospecies level, allowing a detailed comparison between
seamounts and nodule fields.
Three categories of substratum types were distinguished: (1) Predominant soft substrata (<40% hard
substrata), (2) mix or transition (between 40 and 60% hard substrata) and (3) predominant hard
substrata (>60% hard substrata), and were annotated per 10m distance units based on the video
footage and tested for correlations with taxonomic abundances.
ROV transects on the seamounts were carried out as exploratory dives. Sampling strategy both at
seamounts and nodule fields combined video and sampling or specimen collection. Due to varying
altitude of the ROV and the use of camera pan, tilt and zoom, it was not possible to use surface
coverage as a standardisation measure. We used video transect length instead. For the transect
length calculation for each dive, we omitted all parts of the video footage in which the ROV was at
an altitude of >10m, or sections where the ROV was not visualising the seafloor (e.g. during
transiting or inspecting ROV parts or instruments). Visualisation of ancient disturbance tracks were
omitted as well, as these fell out of the scope of the article. Faunal densities were calculated as the
number of observations per 100m, in order to compensate for time spent collecting samples and
differing transect lengths. Statistical testing was carried out in R (R core team, 2018). Non-metric
multidimensional scaling analysis (NMDS) was based on Bray-Curtis dissimilarity and carried out with
the vegan package (Oksanen et al., 2018). The Kendall's coefficient of concordance (W) was
calculated to identify significantly associated groups of species, based on correlations and
permutations (Legendre, 2005).

## 3. Results
About 80% of all taxa observed across the two adjacent ecosystems, could be identified to a
morphospecies level. At a first view, morphospecies revealed to be quite different between
seamounts and nodule fields (Fig. 2). While the number of faunal observations at the seamount
transects were within similar ranges (34-42 ind./100m), those at the nodule transects featured both
highest and lowest values (6.3-67.5 ind/100m) (Table 1). The lowest number of faunal observations
were done at the two APEI3 nodule transects (ROV13 and 14) and highest at the GSR nodule
transect ROV10. What follows is a first description of eastern CCZ seamount megafauna (3.1.) and a
detailed comparison with the neighbouring nodule fields (3.2.)

### 3.1. Insights in CCZ seamount megafauna
The most abundant and diverse (most morphospecies) taxa at the seamount transects comprised
Echinodermata (Asteroidea, Crinoidea, Holothuroidea and Ophiuroidea), Anthozoa (Actiniaria,
Alcyonacea, Pennatulacea, Scleractinia) and Porifera (Hexactinellida) (Table A1, Fig. 3). Keeping in
mind the limitation of the video sampling, differences among the benthic seamount community
composition are described here. The transect at Mann Borgese (APEI3) was characterised by high
densities of Antipatharia, more specifically Antipathidae (18.5 ind./100m), and solitary Scleractinia
(7.9 ind./100m) (Table A1, Fig. 3). Antipathidae observations were mostly grouped at the end of the
video transect, i.e. at the summit. Densities of both Antipatharia and Scleractinia were much lower
on the other seamount transects (<1 ind./100m) with Scleractinia being absent from Heip and
Senckenberg transects. Alcyonacea corals were observed on all seamount transects. Isididae were
found at Senckenberg and Heip transects, and one individual from the Chrysogorgiidae family was
observed at the latter as well. Varying numbers of Primnoidae were observed on all transects (Table
A1). High abundances of Pennatulacea were observed at Senckenberg (3.8 ind./100m), representing
about 28% of sessile fauna annotations for this transect.
Enteropneusta were only observed on Rüppel and Senckenberg transects in the BGR area,
represented by two different morphospecies, namely *Yoda* morphospecies (Torquaratoridae) at
Rüppel and *Saxipendium* morphospecies (Harrimaniidae) at Senckenberg.
Highest Polychaeta densities were observed at Heip transect in the GSR area, which was mainly due
to high densities of free-swimming Acrocirridae (4.2 ind./100m vs. 0. 2ind./100m in BGR area Table
A1). Aphroditidae polychaetes were only present at the BGR transects (0.2 ind./100m
(corresponding to 3 indivuals along the transect) at Rüppel and 0.04 ind/100m (or 1 individual along
the transect) at Senckenberg) (Table A1).
Porifera densities were highest at the Heip transect (7.5 ind./100m), followed by Rüppel
(3.5ind./100m), Senckenberg (1.9ind./100m) and lastly Mann Borgese (1.8ind./100m). Six Porifera
families were annotated featuring >7 to >10 morphospecies (Fig. 3, Table A1). Cladorhizidae (two
individuals) were only observed on Heip transect, and one *Poliopogon* sp. (Pheronematidae) was
observed at Mann Borgese transect.  Rossellidae gen. sp. nov. was present on three seamount
transects, exception being Mann Borgese.
Overall Echinodermata densities were highest at Senckenberg seamount (17 ind./100m), followed by
Rüppel (10 ind./100m) (Table A1, Fig. 3), both adding up to 47% of all image annotations for these
transects. The number of morphospecies for all echinoderm taxa (Asteroidea, Echinoidea,
Holothuroidea and Crinoidea) was also highest at these 2 seamounts in the BGR area (Fig. 3). For
comparison, echinoderms at Heip (10 ind./100m) and Mann Borgese transects (3.3 ind./100m) were
responsible for 32% and 8.2% of observations respectively. Crinoidea densities were highest at
Senckenberg (4.2 ind./100m), while Holothuroidea were most abundant at Rüppel (4.4 ind./100m).
The holothuroid families of Elpidiidae and Laetmogonidae were only observed at Senckenberg and
Rüppel (BGR). Psychropotidae and Synallactidae were observed on all seamounts, represented by
different morphospecies. Deimatidae were not observed on Mann Borgese, but were present on the
three other seamount transects, again with different morphospecies and densities. Velatid
Asteroidea were only observed at Senckenberg and Rüppel (BGR), while Brisingida and Paxillosida
were observed on all four seamounts. Aspidodiadematid Echinoidea were absent from the Heip
transect and urechinid Echinoidea were absent from the Mann Borgese transect.
A species accumulation curve (Fig. 4a) confirmed the limitations of the restricted and exploratory
nature of the sampling as no asymptote was reached. The rarefaction curves (Fig. 4b) showed that
the transects with the most faunal observations, which corresponded here to the longer transects,
were more diverse. However, at smaller sample sizes curves did not cross, thus maintaining the
differences observed at higher sample sizes with the Senckenberg transect (ROV04) as most diverse
followed by Rüppel (ROV02) (both BGR). The video transect carried out at Mann Borgese (ROV15,
APEI3) was the least diverse.
A comparison of all morphospecies observed along the 4 transects was presented in a Venn diagram
(Fig. 5a). Each seamount transect was characterised by a highest number of unique morphospecies,
only observed on the transect in question and not elsewhere. Only three morphospecies were
present in all seamount transects, namely Ceriantharia msp. 2, a small red galatheid crab and a
foliose sponge. Highest number of overlapping morphospecies (#16) was observed between Rüppel
and Senckenberg, both in the BGR area (Fig. 5a). Mann Borgese showed the smallest degree of
overlap with the other transects (Fig. 5a).
About 57% of all sessile fauna was associated with predominantly hard substrata, followed by 31%
on the mixed substrata. For the mobile taxa, the pattern was less pronounced with 41 and 42%
associated with predominantly hard and mixed hard/soft substrata respectively. The amount of
predominantly hard and soft substrata was negatively correlated, though not significantly. This was
due to the equal amounts (40-60%) of mixed hard/soft substrata. Over all seamount transects
pooled together, no taxa were significantly correlated with the amount of hard substrata, nor with
soft substrata. When looking at the individual transects, no significant correlations were found
between taxa and substrata for ROV02 or ROV04 or ROV09, most likely due to the equal distribution
of the amount of hard/soft/mix substrata. In this perspective, ROV15 stood out, as it was dominated
by predominantly hard substrata (56/%). For this transect, Pennatulacea were significantly
negatively correlated with the amount of hard substrata and Zoantharia/Octocorralia were
significantly and positively correlated with hard substrata, as were Ophiuroidea, Asteroidea,
Crinoidea and Mollusca.
Due to the limited sample size, the representativity of the observed biological patterns remains to
be corroborated by a more elaborate sampling strategy.
## 3.2. Comparison of seamount and nodule field faunal composition and variation
The faunal composition and richness (number of morphospecies in higher taxonomic groups) of the
nodule transects can be consulted in Fig. 3 and Table A1, respectively. In concordance with the
seamount transect, the species accumulation curve of the nodule transects did not reach an
asymptote either (Fig. 4c). The rarefaction curves showed that the relations among transects were
less straightforward for the nodule transects versus the seamount ones and did cross at smaller
sample sizes (<100 individuals, Fig. 4d). ROV13 and ROV14 transects (both APEI3) were the longest in
distance travelled (Table 1) but featured less faunal observations. At small sample sizes, the richness
at ROV13 and 14 was highest. ROV08 and ROV10 (both GSR) showed parallel curves with ROV08
being more diverse (Fig. 4d).
A venn diagram showing the morphospecies overlap among the nodule transects showed a total of 5
species re-occurring on all 5 transects (Fig. 5b). These were: Munnopsidae msp. 1 (Isopoda,
Crustacea), Actiniaria msp.7 (Cnidaria), Ophiuroidea msp. 6 (Echinodermata), *Holascus* sp. and
*Hyalonema* sp. (Hexactinellida, Porifera). There was a high number of unique morphospecies for
each transect, though not as high as for the seamount transects (Fig. 5). ROV13 and 14 (both APEI3)
showed least overlap with the other transects, which is similar to what was observed at the
seamounts.
Observations and quantifications of morphospecies confirmed the high degree of dissimilarity
between the two neighbouring ecosystems. Porifera, Ophiuroidea (Echinodermata), Actiniaria and
Alcyonacea (Cnidaria) were more abundant at nodule fields (Fig. 3). These taxonomic groups were
also most diverse on nodule fields (i.e. highest number of morphospecies), exception being the
Alcyonacea which featured more morphospecies on the seamounts (12 to 8 morphospecies for
seamounts and nodule fields respectively) (Fig. 3). Of all Porifera, Cladhorizidae were more diverse
at nodule fields than at seamounts (7 to 1 morphospecies, respectively).
There were only 21 morphospecies (10%) that were observed both on seamounts and nodule fields
(Fig. 6). While this subset of morphospecies occurred in both ecosystems, they did so in very
different densities, i.e. very abundant in one ecosystem and very low in abundance in the other:
examples are Galatheidae small red msp. (Decapoda, Crustacea), *Synallactes* white msp.
(Holothuroidea), Ophiuroidea msp. 5 and 6, Comatulida msp. 1 (Crinoidea), *Hyalonema* sp. and
*Hyalostylus* sp. (both Hexactinellida, Porifera) (Fig. 6).
Three Ophiuroidea morphospecies were present at both seamounts and nodule fields (Fig. 2, 3 and
6). The majority of the very abundant Ophiuroidea observed at the CCZ seamounts were small and
situated on hard substrata (morphospecies 5), while most of the Ophiuroidea at nodule fields
(including morphospecies 6) were observed on the soft sediments. Morphospecies 6 was only rarely
observed on the seamounts (Fig. 3). Another easily recognisable morphospecies was found on
Porifera, coral and animal stalks and was more abundant at seamounts than at nodule fields
(morphospecies 4) (Fig. 2 and 3).
Crinoidea, Asteroidea (both Echinodermata) and Antipatharia (Cnidaria) were more abundant on the
seamounts (Fig. A1). This coincided with a higher diversity for Asteroidea and Antipatharia on the
seamounts as well. Crinoidea diversity was similar (5 to 4 morphospecies comparing seamounts to
nodule fields). Holothuroidea occurred in similar densities in both ecosystems (Fig. A1, though they
were characterised by different morphospecies (Fig. 3). Overall densities of Echinoidea were
comparable between seamounts and nodule fields, though for the nodule fields this was mostly due
to one very abundant morphospecies, namely Aspidodiadematidae msp 1, which was absent at the
seamounts (Fig. 3). Besides this, Echinoidea were more diverse at seamounts (11 morphospecies vs.
5 at nodule fields).
There was no morphospecies overlap for Tunicata, Antipatharia, and Actiniaria. Alcyonacea,
Ceriantharia, Corallimorphidae and Crinoidea only shared 1 morphospecies between seamounts and
nodule fields, namely *Callozostron* cf. *bayeri*, Ceriantharia msp. 2, *Corallimorphus* msp. 2 and
Comatulida msp. 1 respectively (Fig. 6).
There were no observations of Enteropneusta, Scleractinia and Zoantharia (Cnidaria), Aphroditidae
(Polychaeta) or holothuroid Deimatidae at the nodule fields transects (Table A1, Fig. A1). While
Actinopterygii were left out of the analysis, it should be noted that fish observations were more
diverse at the seamounts than on the nodule fields.
There was quite some faunal variation observed among the video transects of, both seamounts and
nodule fields (see Fig. 5). The (dis)similarities were analysed by a nMDS analysis, which grouped the
9 different video transects based on their taxonomic composition. Despite the large intra-ecosystem
variation, they pooled in two distinct groups separating the nodule fields from the seamounts (Fig.
7a). Within each group, BSR and GSR transects were more similar to one another both for
seamounts and nodule fields, whilst APEI3 transects stood out more.
The Kendall's coefficient of concordance (W, Legendre, 2005) corroborated the existence of two
significantly different species associations, whose composition corresponded to the fauna
characterising the nodule fields (W=0.20, p<0.001, after 999 permutations) and the seamounts
(W=0.30, p<0.001, after 999 permutations).
Depth was fitted as a vector on top of the nMDS plot (Fig. 7b) and showed that the discrepancy in
faunal composition between the two ecosystems also corresponded to a difference in depth, with
the nodule transects all being situated below the 4000m isobath and the seamount transects ranging
from 1650 to >3500m (Fig. 7b).

## 4. Discussion

### 4.1. Intra-ecosystem faunal variation

Community composition varied markedly at seamounts and nodule fields. The limited sampling (n=9
transects), at different locations and additionally, for the seamounts, different depth ranges,
precluded any general conclusions on quantifications of biodiversity *per se*. However, taking this into
account, it was also the first time seamounts were visited in the area, thus granting first insights in
the fauna inhabiting these seamounts and allowing a first comparison with nodule faunal
composition.
The two BGR seamount transects were most similar in faunal composition, followed by the Heip
seamount transect (GSR). These seamount video transects were characterised by more similar depth
ranges, and the two BGR transects were also geographically closest to each other. Although for
seamounts, distance separating them might be a less determining factor than depth since
(mega)faunal communities can be very different even between adjacent seamounts (Schlacher et al.,
2014; Boschen et al., 2015). Overall, parameters that vary with depth, such as temperature, oxygen
concentration, substratum type, food availability, and pressure are considered major drivers of
species composition on seamounts (Clark et al., 2010; McClain et al., 2010). The quantification of the
amount of hard and soft substrata was not distinctive enough to explain differences observed here.
The difference in depth could also explain the higher dissimilarity with Mann Borgese (APEI3) who
featured the shallowest transect and summit, which was dominated by Antipatharia. Antipatharians
were previously reported to be more dominant towards peaks as compared to mid-slopes at
corresponding depths (Genin et al., 1986). Based on their filter-feeding strategy, Porifera (except
carnivorous Cladorhizidae), were also thought to benefit from elevated topography (peaks) or
exposed substrata in analogy to corals (Genin et al., 1986; Clark et al., 2010), though no such pattern
was apparent here. Porifera are notoriously difficult to identify based on imagery. Although the
sampled individuals allowed some identifications to genus or species level (Kersken et al., 2018a and
b), identifications remained hard to extrapolate across the different video transects. Generally, as in
our study, seamount summits have been more intensively sampled (Stocks, 2009) although the little
work done at seamount bases and deep slopes indicated that these areas support distinct
assemblages (Baco, 2007).
Among the nodule transects a considerable amount of variation in faunal composition was observed
(this study, Vanreusel et al., 2016). The two APEI3 nodule transects (ROV13 and 14) stood out in
faunal composition, diversity and in low number of faunal observations. They were also the only two
transects situated below the 4500m isobaths. But rather than depth, the nodule coverage may be
considered an important driving factor, since the density of nodule megafauna was shown to vary
with nodule size and density/coverage (Stoyanova, 2012; Vanreusel et al., 2016, Simon-Llédo et al.,
2019). Here as well, the APEI3 transects were characterised by a high nodule coverage (~40-88%,
Vanreusel et al., 2016), whereas the BGR and GSR nodule transects (ROV3 and ROV 8 + 10,
respectively) had a nodule coverage <30% and were also more similar in faunal composition
(Vanreusel et al., 2016). Other factors that could be at play are the more oligotrophic surface waters
of the northern CCZ which could be the cause of the overall lower faunal densities at APEI3 nodule
fields (Vanreusel et al., 2016). Volz et al. (2018) corroborated this, with the location of the APEI3 site
in the proximity of the carbon-starved North Pacific gyre being characterised by a reduced POC-flux
quantified to being 22-46% lower than the GSR and BGR areas respectively.
The species accumulation curves showed that no asymptote was reached neither at seamounts, nor
at nodule fields. Consequently, longer transect lengths might be necessary to representatively
quantify and assess megafauna density and diversity (Simon-Lledó et al., 2019). In addition, for a first
in-depth description and assessment of seamount fauna composition, one video transect is
insufficient to describe the diversity and shifts in faunal assemblages of the surveyed seamounts.
Rather, an ampler imaging strategy should be developed, with a minimum transect length exceeding
1000ms (Simon-Llédo et al., 2019) and replicate transects carried out on different faces of the
seamount, on slopes with varying degree of exposure to currents and different substrate types.
Wider depth ranges should be taken into account as well. Alternatively, across slope transects,
following depth contours should be considered as these could provide observation replicates for a
given depth. Despite its limitations, this study grants first insights in the seamount inhabiting
megafauna of the eastern CCZ and an important first comparison with nodule fauna.
## 4.2. Faunal (dis)similarities between seamounts and nodule fields
In other areas, seamounts were shown to share fauna with surrounding habitats (Clark et al., 2010)
and could thus potentially serve as source populations for neighbouring environments (McClain et
al., 2009). While generally few species seemed restricted to seamounts only (Clark et al., 2010),
morphospecies in this study revealed to be quite different between seamounts and nodule fields
with little overlap between both. Despite the high degree of variation observed among all the video
transects, these grouped into two distinctly separate clusters, separating nodule from seamount
transects. The few overlapping morphospecies did occur in different densities in each ecosystem,
implying a different role or importance in the ecological community and its functioning.
Overall, nodule fields showed higher faunal densities than seamounts. Shifts in density patterns
between nodule fields and seamounts were more evident in a number of taxa, where the variety of
morphospecies and feeding strategy within each group was likely to be at play. An example of this
are the Echinodermata, which include Asteroidea (predators and filter feeders (Brisingida)),
Crinoidea (filter feeders), Echinoidea (deposit feeders), Holothuroidea (deposit feeders) and
Ophiuroidea (omnivores). Asteroidea were more abundant on seamounts and both Echinoidea and
Asteroidea were more diverse in this ecosystem as well. Ophiuroidea were most abundant on the
nodule fields (ratio 7 to 1 when compared to seamounts). Same ophiuroid morphospecies were
present at seamounts and nodule fields but in very different abundances and they were observed on
different substrata types, which indicates different lifestyles, feeding behaviour and corresponding
dietary specialisations (Persons and Gage, 1984). Previously it was already demonstrated that
Ophiuroidea did not show high levels of richness or endemism on seamounts (O'Hara, 2007). At
nodule fields, Ophiuroidea were often observed in association with xenophyophores (Amon et al.,
2016, this study) and a similar observation was done at east Pacific seamounts off Mexico (Levin et
al., 1986), though no such associations were observed on the seamounts studied here.
Holothuroidea densities were thought to possibly decrease when less soft sediment was available
since they feed mainly on the upper layers of the soft-bottom sediment (Bluhm and Gebruk, 1999).
No significant link was established between holothuroid densities and the amount of hard substrata
in this study, but their community composition varied distinctly between nodule fields and
seamounts with more families being observed at the latter. Additionally, at the seamounts, many
holothurians were observed on top of rocks, possibly reflecting different feeding strategies and
explaining the observations of different morphospecies. Geographical variations, different bottom
topography, differences in nodule coverages and sizes and/or an uneven distribution of holothurians
on the sea floor were thought to play a role in holothuroid community composition (Bluhm and
Gebruk, 1999). On the other hand, variability in deep-sea holothuroid abundance was proposed to
depend primarily on depth and distance from continents (see Billet, 1991 for a review).
Stalked organisms, such as Crinoidea (Echinodermata) and Hexactinellida (except for
Amphidiscophora, Porifera) rely on hard substrata for their attachment and are considered being
among the most vulnerable organisms when mining is concerned. Crinoidea were more abundant on
seamounts, possibly because hard substrata were less limiting than in the nodule fields. Porifera
densities (stalked and non-stalked) varied among all analysed transects, revealing no particular
trends in abundance. However, the species composition of deep-sea glass sponge communities from
seamounts and polymetallic nodule fields was distinctly different. Polymetallic nodule field
communities were dominated by widely-distributed genera such as *Caulophacus* and *Hyalonema*,
whereas seamount communities seemed to have a rather unique composition represented by
genera like *Saccocalyx*.
Corals were generally considered to be more abundant on seamounts than adjacent areas, due to
their ability to feed on a variety of planktonic or detritus sources suspended in the water column
(Rowden et al., 2010). In this study, the Alcyonacea densities were lower on the seamounts than on
the nodule transects. The majority of Alcyonacea morphospecies of the seamounts did not occur on
the nodule fields and vice versa, with exception of *Callozostron* cf. *bayeri* which was also present at
the nodule fields but in very low densities (1/10 of those observed at seamounts). The Antipatharia
were most abundant at the Mann Borgese seamount (APEI3) compared to all other transects. The
depth difference of more than 3000m between this particular seamount and the nodule fields could
explain the abundance in Antipatharia which were shown to be more abundant at lower depths
(Genin et al., 1986). Additional presence of Pennatulacea at seamounts, a taxon that was virtually
absent from the nodule field transects and that appeared more linked to predominant soft substrata
at seamounts, resulted in completely distinct coral communities for each ecosystem.
Actiniaria were denominated the second most common group at CCZ nodule fields, after the
xenophyophores (Kamenskaya et al., 2015) and, in our study, were also more abundant on nodule
fields than on seamounts. Depending on the species and feeding strategy, the ratio hard/soft
substrata and their preference for either one could play a role. Since morphospecies were distinct
between seamounts and nodule fields, their role in the respective communities are likely to differ as
well. Combinations of deposit feeding and predatory behaviour in Actiniaria have been observed, as
well as burrowing activity, preference for attachment to hard substrata and exposure to currents
(Durden et al., 2015a; Lampitt and Paterson, 1987; Riemann-Zürneck, 1998), all factors that could
influence the differences in morphospecies observed.
Some taxa were only observed on the seamounts in this study, while they occurred on nodule fields
elsewhere, be it in low densities. For instance, Enteropneusta, which in this study were found only
on seamounts, were observed previously at CCZ nodule fields though observations were rather rare
(Tilot, 2006). They appeared more abundant at the nodule fields of the Deep Peru Basin (DISCOL
area), though a wide range in abundances was displayed there as well (Bluhm, 2001). The exception
were the Scleractinia, which were quite common on seamounts, as also reported in other studies
(e.g.Baco, 2007, Rowden et al., 2010), but distinctly absent at nodule fields.
Explanation for the discrepancies in faunal composition and the low degree of morphospecies
overlap between seamount and nodule fields, as observed here, can be multiple. For one, nodules
may not be considered a plain hard substratum, with their metal composition, microbial colonisation
and the nodule/sediment interface influencing the epi-and associated megafaunal composition. The
possibility of a specific deep-sea faunal community that tolerates or benefits from manganese
substrata has been previously proposed (Mullineaux, 1988). The comparison between seamounts
and nodule fields as two neighbouring hard-substrata ecosystems also entailed a comparison
between depth gradients and possible thresholds (>4000m for nodule fields and 1500>x <4000m for
seamounts). Related to this is the steepness of the seamount slope and its current exposure playing
a role in the faunal colonisation (Genin et al., 1986; Rappaport et al., 1997).  Other studies showed
that habitat heterogeneity increased megafaunal diversity at seamounts (Raymore, 1982) and
elsewhere, such as abyssal plains (Lapointe and Bourget, 1999; Durden et al., 2015b, Leitner et al.,
2017, Simon-Llédo et al., 2019). Within this perspective the smaller-scale substratum heterogeneity
transcending the ratio hard/soft substrata or amount of hard substrata could play a role as well.

## 5. Conclusions

Based on our current knowledge; seamounts appear inadequate as refuge areas to help maintain
nodule biodiversity. In order to conclusively exclude seamount habitats as a refuge for nodule fauna,
a more comprehensive sampling should be carried out. The sampling strategy wielded in this study
lacked replicates, uniformity and was limited in sample size. Seamount bases should be taken into
consideration as well as they can be characterised by distinctly different assemblages than the
summits and they feature depth ranges more similar to nodule fields.
While their role as refuge area for nodule field fauna is currently debatable, the possible uniqueness
of the seamount habitat and its inhabiting fauna implies that seamounts need to be included in
management plans for the conservation of the biodiversity and ecosystems of the CCZ.

## Author Contributions

DC, PAR, SPR, DK analysed the images. DC analysed the data. PMA, PAR, AC conceptualised and
carried out the sampling. All authors contributed to the redaction of the manuscript.

## Data Availability

Data sets are made available through OSIS-Kiel data portal, BIIGLE and PANGAEA.

## Competing interest

The authors declare that they have no conflict of interest

## Acknowledgments

We thank the crew of SO239 and GEOMAR for their support in acquiring the images used in this article. The EcoResponse cruise with RV Sonne was financed by the German Ministry of Education and Science BMBF as a contribution to the European project JPI-Oceans "Ecological Aspects of Deep-Sea Mining". This study had the support of PO AÇORES 2020 project Acores-01-0145-Feder-000054_RECO and of Fundação para a Ciência e Tecnologia (FCT), through the strategic projects UID/MAR/04292/2013 granted to MARE. The authors acknowledge funding from the JPI Oceans— Ecological Aspects of Deep Sea Mining project by Fundação para a Ciência e Tecnologia de Portugal (Mining2/0005/2017) and the European Union Seventh Framework Programme (FP7/2007–2013) under the MIDAS project, grant agreement n° 603418. DC is supported by a post-doctoral scholarship (SFRH/BPD/110278/2015) from FCT. PAR was funded by the Portuguese Foundation for Science and Technology (FCT), through a postdoctoral grant (ref. SFRH/BPD/69232/2010) funded through QREN and COMPETE. SPR is supported by FCT in the scope of the "CEEC Individual 2017" contract (CEECIND/00758/2017) and CESAM funds (UID/AMB/50017/2019) through FCT/MCTES. AC is supported by Program Investigador (IF/00029/2014/CP1230/CT0002) from FCT. PMA acknowledges funding from BMBF contract 03 F0707E. Pictures were provided by GEOMAR (Kiel).

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

# Tables

Table 1: Overview table on details of imagery transects analysed in the Clarion-Clipperton license
areas. Video duration includes time spent sampling. Number of observations include undetermined
organisms. Transect lengths do not include parts visualising ancient disturbance tracks or parts when
the seafloor was not visualised or visible.

| Station/Dive | License Area | Seamount (SM) or Nodule field (NF) | Depth (m) | Video duration | Transect length | # obs/ dive | # obs /100m |
|---|---|---|---|---|---|---|---|
| SO239_29_ROV02 | BGR | SM: Rüppell | 3000-2500 | 7h47 | 1250m | 429 | 34.3 |
| S0239_41_ROV03 | BGR | NF | 4080-4110 | 6h32 | 1590m | 932 | 58.6 |
| SO239_54_ROV04 | BGR | SM: Senckenberg | 3350-2850 | 8h45 | 2500m | 890 | 35.6 |
| S0239_131_ROV08 | GSR | NF | 4470-4480 | 7h35 | 710m | 445 | 62.8 |
| SO239_135_ROV09 | GSR | SM: Heip | 3900-3550 | 7h35 | 1000m | 359 | 35.9 |
| S0239_141_ROV10 | GSR | NF | 4455-4480 | 7h35 | 520m | 351 | 67.5 |
| S0239_189_ROV13 | APEI 3 | NF | 4890-4930 | 9h01 | 1790m | 113 | 6.3 |
| S0239_200_ROV14 | APEI 3 | NF | 4650-4670 | 9h19 | 1490m | 179 | 12.0 |
| SO239_212_ROV15 | APEI 3 | SM: Mann Borgese | 1850-1650 | 6h25 | 900m | 378 | 42.0 |










# Figures

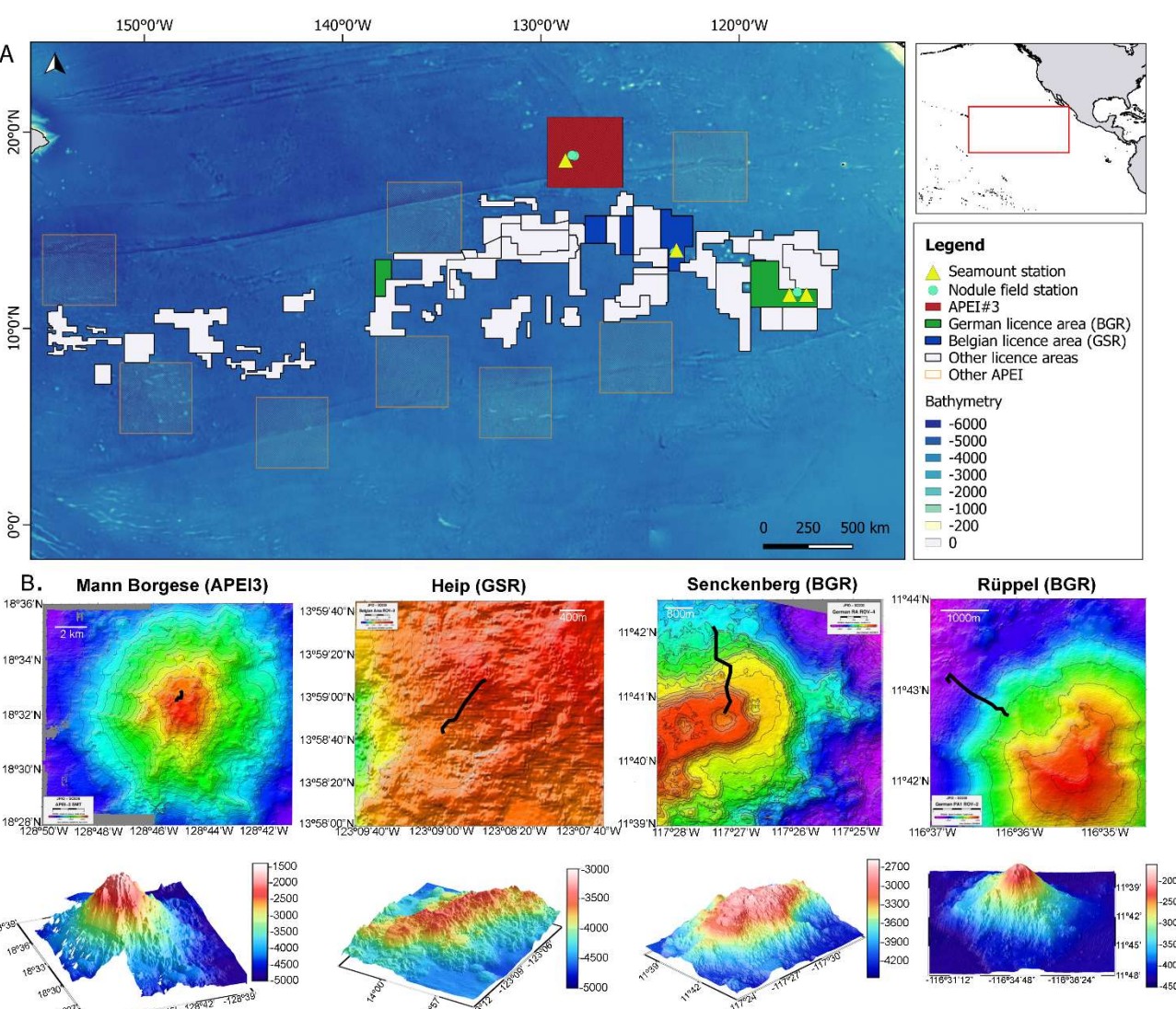

Fig. 1. (A). Location of the Clarion-Clipperton Fracture zone in the equatorial eastern Pacific Ocean
featuring the contract areas from the International Seabed Authority (ISA) and the positions of the
sampled areas (seamounts and nodule fields). Information on transect length and depth gradients
can be found in Table 1. (B). Location of the seamount transects carried out towards the summit on
the north –north-western flank and seamount profiles. Rüppel (BGR, ROV02) and Mann Borgese
(APEI3, ROV15) are single seamounts, while Senckenberg (BGR, ROV04) and Heip (GSR, ROV09) are
sea-mountain ranges.


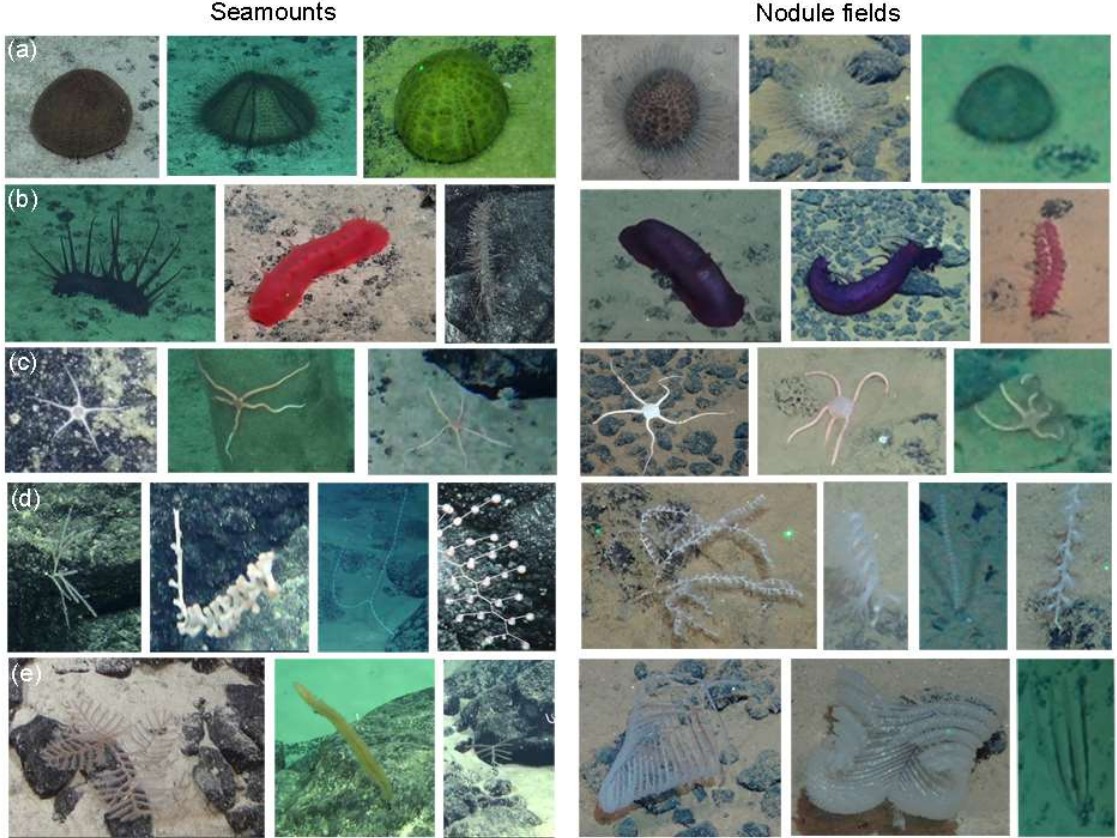

Fig. 2. Some examples of different morphospecies at seamounts and nodule fields in the CCZ.
Selected taxa were (a) Echinoidea (from left to right, Urechinidae msp 4 (URC_019), Urechinidae msp
2 (URC_013), Urechinidae msp 3 (URC_009), Urechinidae msp. A (URC_020), Urechinidae msp. B
(URC_021), Urechinidae msp. C (URC_005), (b) Holothuroidea (from left to right, Psychropotidae
msp 1 (HOL_088), *Benthodytes* red msp. (HOL_101), Deimatidae - irregular papillae msp. (HOL_070),
*Psychropotes verrucosa* (HOL_045), Laetmogonidae (HOL_030), *Synallactes* msp 2 pink (HOL_008)(c)
Ophiuroidea (from left to right, Ophiuroidea msp. 5 (OPH_003), Ophiuroidea msp. 4 (OPH_005),
Ophiuroidea msp. 6 (OPH_006), Ophiuroidea msp. 6 (OPH_006), Ophiuroidea (OPH_012),
Ophiuroidea msp. 4 (OPH_005)), (d) Alcyonacea (from left to right, *Callozostron* cf. *bayeri (*ALC_009),
*Bathygorgia* aff. *profunda* 2 (ALC_005), *Keratoisis* aff. *flexibilis* msp 2 (ALC_029), *Chrysogorgia* cf.
*pinnata*, *Abyssoprimnoa* cf. *gemina* (ALC_008), *Bathygorgia* aff. *profunda* 1, *Calyptrophora* cf.
*persephone* (ALC_007), *Bathygorgia* aff. *abyssicola* 1 (ALC_003), (e) Antipatharia (*Umbellapathes* aff.
*helioanthes* (ANT_018), cf. *Parantipathes* morphotype 1 (ANT_017), *Bathypates* cf. *alternata* msp 1
(ANT_010), *Bathypathes* cf. *alternata* (ANT_006), *Abyssopathes* cf. lyra (ANT_022), *Bathypathes* sp.
(ANT_003)). Codes refer to an ongoing collaboration in creating one species catalogue for the CCZ
and align all morphospecies of different research groups. Copyright: SO239, ROV Kiel 6000, GEOMAR
Helmholtz Centre for Ocean Research Kiel

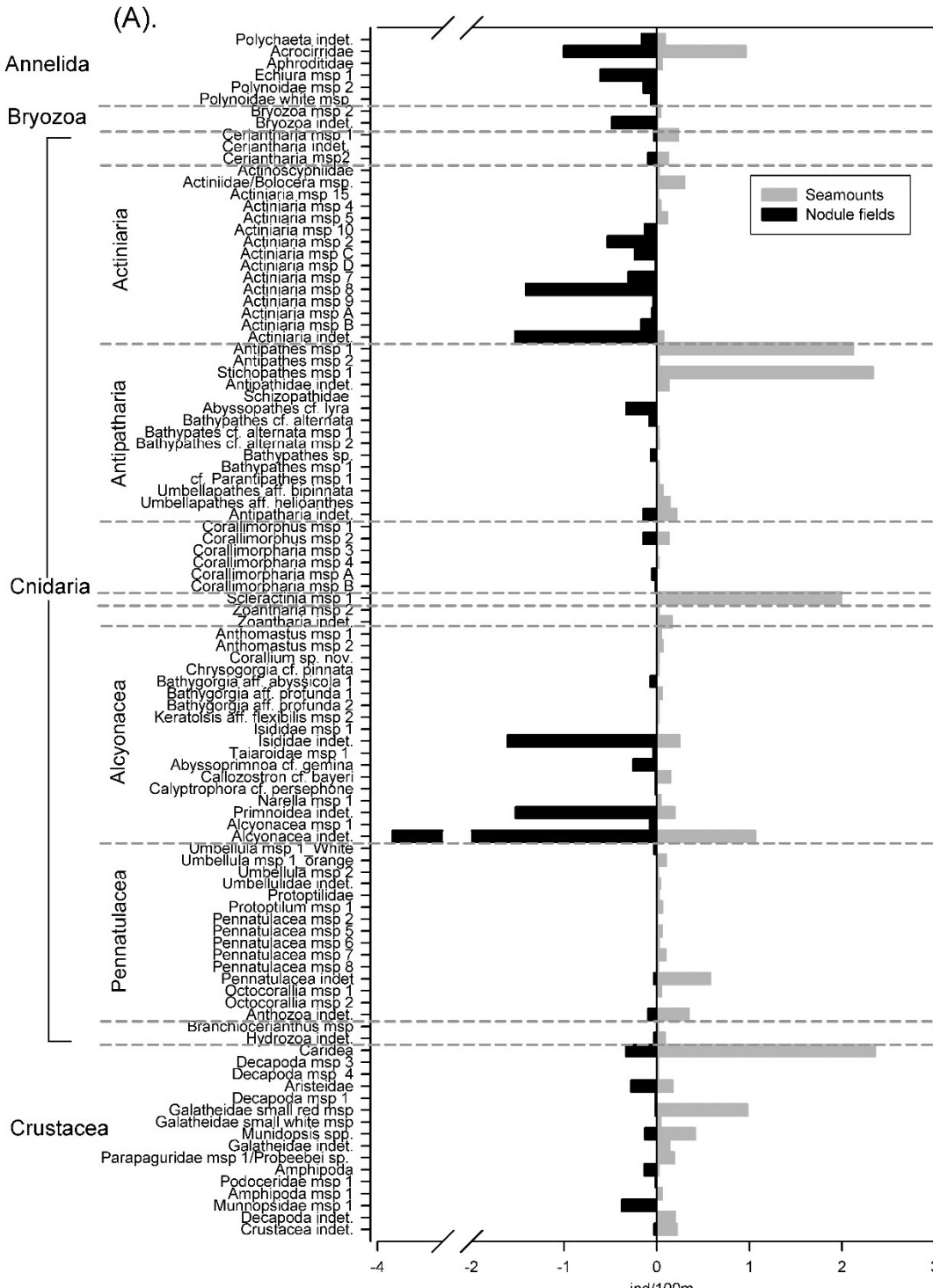


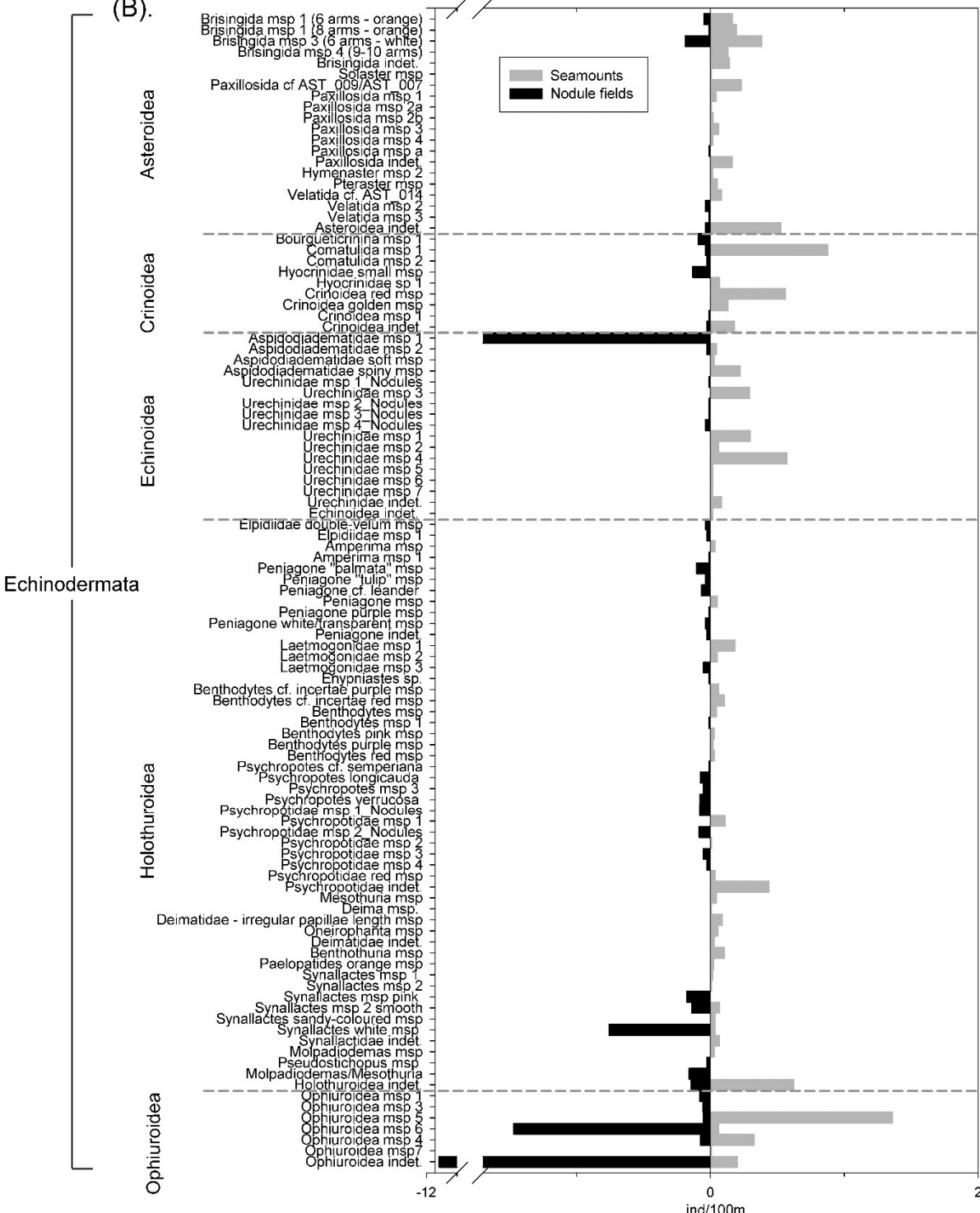

(B).

Seamounts
Nodule fields

Echinodermata

Asteroidea

Brisingida msp 1 (6 arms - orange)
Brisingida msp 1 (8 arms - orange)
Brisingida msp 3 (6 arms - white)
Brisingida msp 4 (9-10 arms)
Brisingida indet.
Solaster msp
Paxillosida cf AST_009/AST_007
Paxillosida msp 1
Paxillosida msp 2a
Paxillosida msp 2b
Paxillosida msp 3
Paxillosida msp 4
Paxillosida msp a
Paxillosida indet.
Hymenaster msp 2
Pteraster msp
Velatida cf. AST_014
Velatida msp 2
Velatida msp 3
Asteroidea indet.

Crinoidea

Bourgueticrinina msp 1
Comatulida msp 1
Comatulida msp 2
Hyocrinidae small msp
Hyocrinidae sp 1
Crinoidea red msp
Crinoidea golden msp
Crinoidea msp 1
Crinoidea indet.

Echinoidea

Aspidodiadematidae msp 1
Aspidodiadematidae msp 2
Aspidodiadematidae soft msp
Aspidodiadematidae spiny msp
Urechinidae msp 1_Nodules
Urechinidae msp 3
Urechinidae msp 2_Nodules
Urechinidae msp 3_Nodules
Urechinidae msp 4_Nodules
Urechinidae msp 1
Urechinidae msp 2
Urechinidae msp 4
Urechinidae msp 5
Urechinidae msp 6
Urechinidae msp 7
Urechinidae indet.
Echinoidea indet.

Holothuroidea

Elpidiidae double-velum msp
Elpidiidae msp 1
Amperima msp
Amperima msp 1
Peniagone "palmata" msp
Peniagone "tulip" msp
Peniagone cf. leander
Peniagone msp
Peniagone purple msp
Peniagone white/transparent msp
Peniagone indet.
Laetmogonidae msp 1
Laetmogonidae msp 2
Laetmogonidae msp 3
Enypniastes sp.
Benthodytes cf. incertae purple msp
Benthodytes cf. incertae red msp
Benthodytes msp
Benthodytes msp 1
Benthodytes pink msp
Benthodytes purple msp
Benthodytes red msp
Psychropotes cf. semperiana
Psychropotes longicauda
Psychropotes msp 3
Psychropotes verrucosa
Psychropotidae msp 1_Nodules
Psychropotidae msp 1
Psychropotidae msp 2_Nodules
Psychropotidae msp 2
Psychropotidae msp 3
Psychropotidae msp 4
Psychropotidae red msp
Psychropotidae indet.
Mesothuria msp
Deima msp.
Deimatidae - irregular papillae length msp
Oneirophanta msp
Deimatidae indet.
Benthothuria msp
Paelopatides orange msp
Synallactes msp 1
Synallactes msp 2
Synallactes msp pink
Synallactes msp 2 smooth
Synallactes sandy-coloured msp
Synallactes white msp
Synallactidae indet.
Molpadiodemas msp
Pseudostichopus msp
Molpadiodemas/Mesothuria
Holothuroidea indet.

Ophiuroidea

Ophiuroidea msp 1
Ophiuroidea msp 3
Ophiuroidea msp 5
Ophiuroidea msp 6
Ophiuroidea msp 4
Ophiuroidea msp7
Ophiuroidea indet.

-12                                    0                                    2

ind/100m


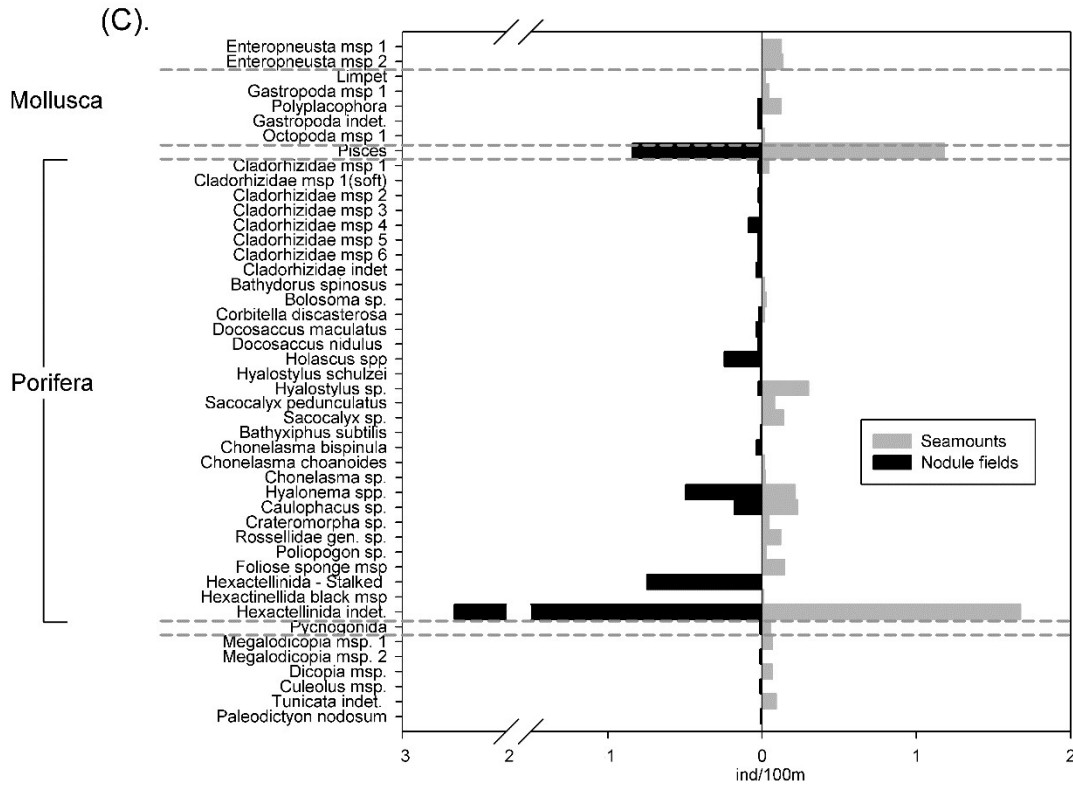


Fig. 3. Back-to-back histogram comparing average densities of morphospecies and taxa (ind/100m)
for seamount (#4) and nodule field (#5) video transects. (a) Annelida, Bryozoa, Cnidaria and
Crustacea, (B) Echinodermata and (C) Mollusca, Porifera, Hemichordata and Chordata (Tunicata).



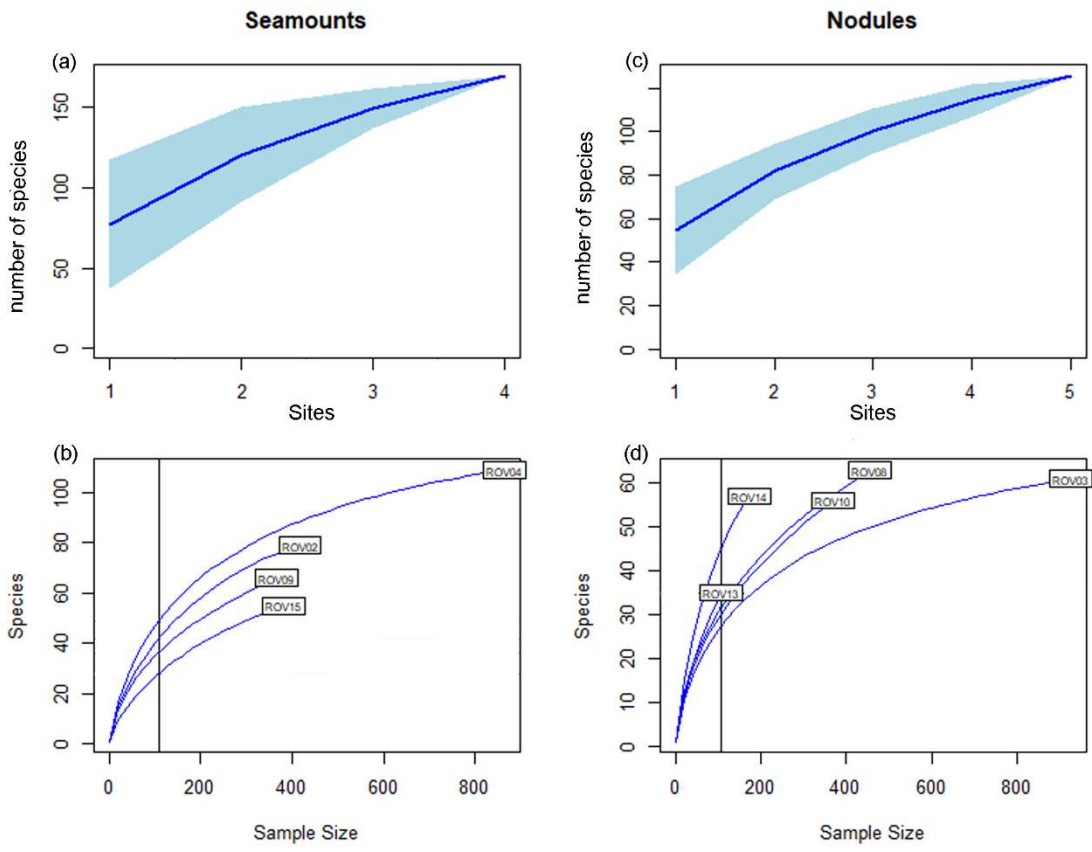

Fig. 4. Species accumulation (upper panel, a and c) and rarefaction curves (lower panel, b and d) for
the seamount (#4) and nodule field (#5) transects. Seamount dives: ROV02= Rüppel (BGR),
ROV04=Senckenberg (BGR), ROV09=Heip (GSR), ROV15=Mann Borgese (APEI3) in the lower left
panel (b). Nodule field dives: ROV03 was carried out in the BGR area, ROV08 and 10 in the GSR area
and ROV13 and 14 in the APEI3, presented in the lower right panel (d). Sample size is the number of
individuals. Vertical line in the lower panel shows sample size=100.



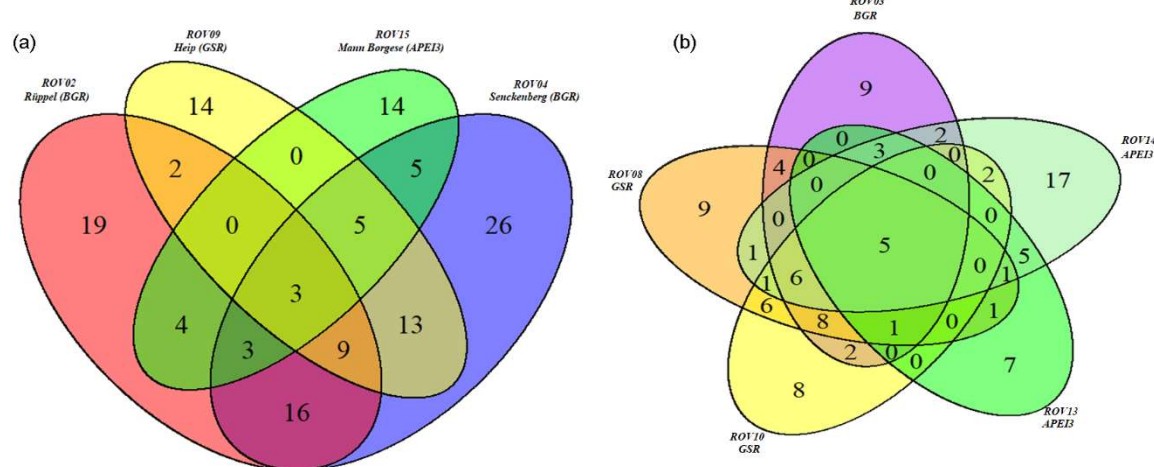

Fig. 5. A Venn diagram showing the unique and shared morphospecies among seamount video
transects. Values are indicative rather than absolute due to different transect lengths and
differences in richness. Left panel (a) features seamount transects and the right panel features the 5
nodule field transects. Colour codes were adapted among panels, with APEI3 nodule transects in
green, related to Mann Borgese seamount transect. BGR (ROV03) transect was purple in
correspondence to BGR seamount transects (red=Rüppel and blue=Senckenberg). GSR transects
(ROV08 and 09) were shades of yellow.





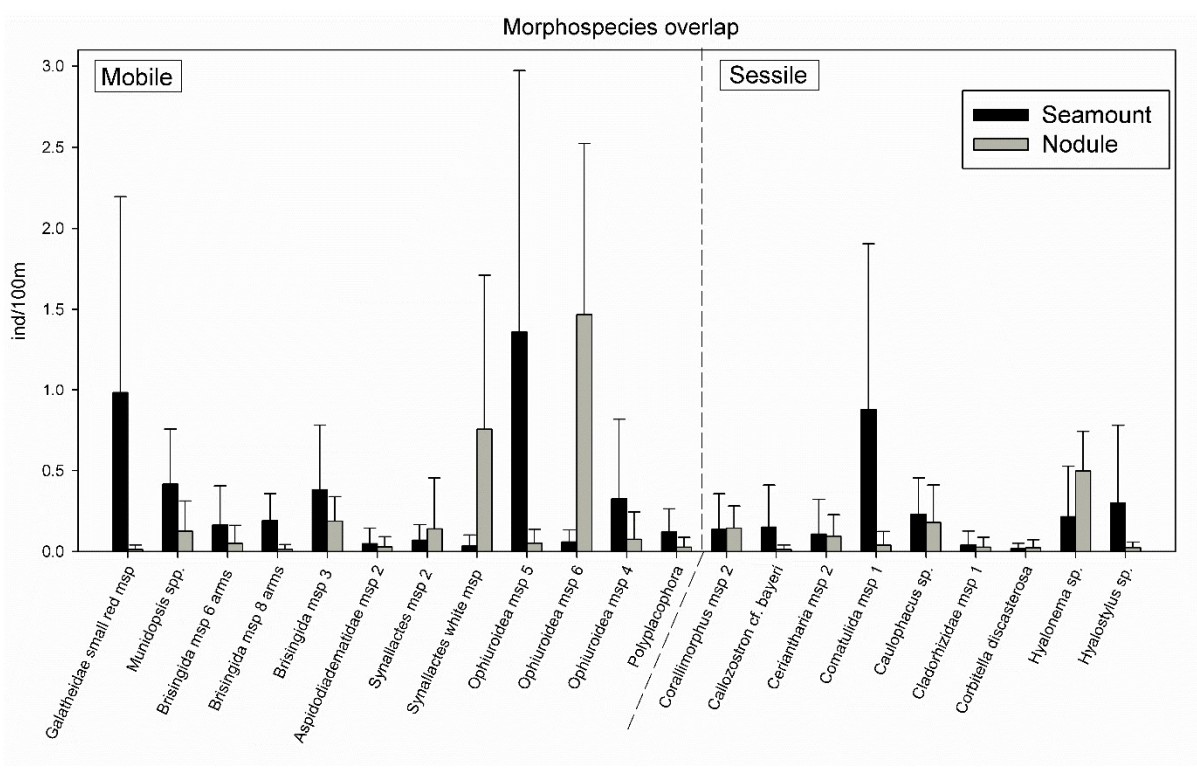


Fig. 6. Morphospecies present in both seamounts and nodule field transects and their average
density (ind/100m) and standard deviation per ecosystem.

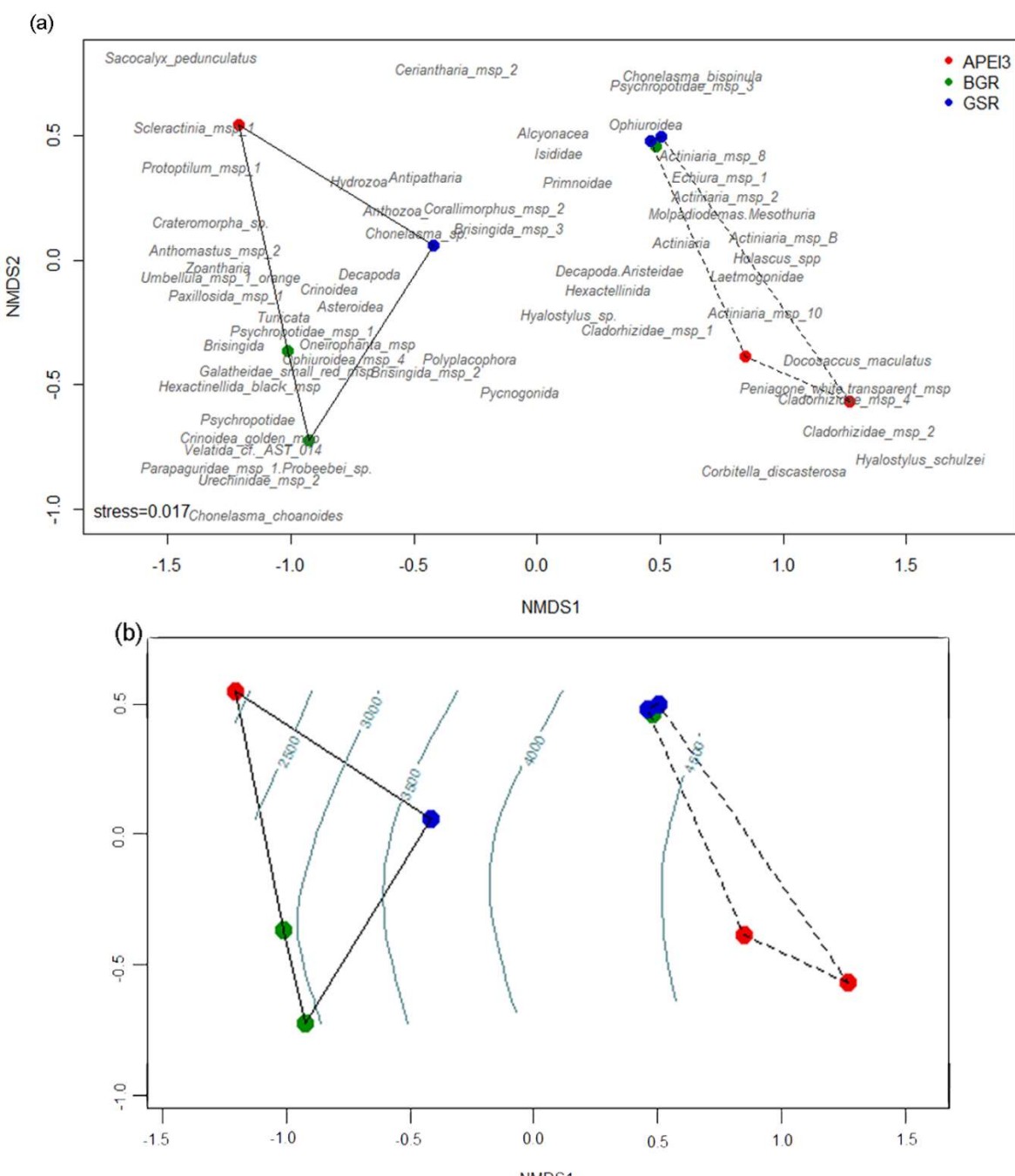

Fig. 7. nMDS-plot with faunal densities and Bray-Curtis distances. Upper panel (a) presents the
grouping of the video transects based on their faunal composition and lower panel (b) features the
same plot but with depth as a vector fitting. Dotted lines group the nodule transects while the full
line groups the seamount transects.




Table A1. Overview of all densities (ind./100m) observed within each video transect. Higher taxa are
in bold. * indicates taxa left out of the statistical analyses due to lack of representativity. Indets were
organisms impossible to attribute to a lower taxonomic group. ROV02=Rüppel, ROV04=Senkcenberg,
ROV09=Heip, ROV15=Mann Borgese

| | SEAMOUNTS | | | | NODULE FIELDS | | | | |
| --- | --- | --- | --- | --- | --- | --- | --- | --- | --- |
| | ROV2 ind/100m | ROV4 ind/100m | ROV9 ind/100m | ROV15 ind/100m | ROV3 ind/100m | ROV8 ind/100m | ROV10 ind/100m | ROV13 ind/100m | ROV14 ind/100m |
| **Annelida*** | | | | | | | | | |
| **Polychaeta indet.** * (No Serpulidae) | 0.14 | 0.12 | | 0.11 | 0.31 | | 0.38 | 0.06 | 0.07 |
| Acrocirridae | 0.14 | 0.16 | 3.56 | | 0.57 | 0.14 | 0.58 | 1.79 | 1.95 |
| Aphroditidae | 0.20 | 0.04 | | | | | | | |
| Echiura msp 1 | | | | | 0.57 | 1.13 | 1.15 | | 0.20 |
| Polynoidea | | | | | | | | | |
| Polynoidae msp 2 | | | | | | 0.14 | 0.58 | | |
| Polynoidae white msp | | | | | | 0.14 | | 0.06 | 0.13 |
| | | | | | | | | | |
| **Bryozoa** | | | | | | | | | |
| Bryozoa msp 2 | | | 0.17 | | | | | | |
| Bryozoa indet. | | 0.038 | | | 0.44 | 1.55 | 0.19 | 0.11 | 0.13 |
| | | | | | | | | | |
| **Cnidaria** | | | | | | | | | |
| *Anthozoa* | | | | | | | | | |
| Ceriantharia | | | | | | | | | |
| Ceriantharia msp 1 | 0.34 | 0.04 | 0.34 | 0.22 | | | | | |
| Ceriantharia msp 2 | | | | 0.43 | | 0.28 | 0.19 | | |
| Ceriantharia indet. | | | 0.08 | | | | | | |
| **Hexacorralia** | | | | | | | | | |
| Actiniaria | | | | | | | | | |
| Actinoscyphiidae | | 0.12 | | | | | | | |
| Actiniidae/*Bolocera* msp. | 1.02 | 0.19 | | | | | | | |
| Actiniaria msp 15 | 0.07 | | | | | | | | |
| Actiniaria msp 4 | | 0.08 | | 0.11 | | | | | |
| Actiniaria msp 5 | 0.07 | 0.08 | | 0.32 | | | | | |

| | C1 | C2 | C3 | C4 | C5 | C6 | C7 | C8 | C9 |
|---|---|---|---|---|---|---|---|---|---|
| Actiniaria msp 10 | | | | | 0.31 | | | | 0.34 |
| Actiniaria msp 2 | | | | | 1.07 | 1.13 | 0.19 | | 0.27 |
| Actiniaria msp C | | | | | 0.38 | 0.42 | 0.38 | | |
| Actiniaria msp D | | | | | 0.06 | | | | |
| Actiniaria msp 7 | | | | | 0.63 | 0.14 | 0.58 | 0.06 | 0.13 |
| Actiniaria msp 8 | | | | | 0.13 | 3.66 | 3.08 | | 0.20 |
| Actiniaria msp 9 | | | | | | | 0.19 | | |
| Actiniaria msp A | | | | | | | 0.19 | | 0.07 |
| Actiniaria msp B | | | | | 0.25 | 0.14 | 0.38 | 0.06 | |
| Actiniaria indet. | 0.14 | 0.15 | | | 1.57 | 1.41 | 4.42 | 0.11 | 0.13 |
| Antipatharia | | | | | | | | | |
| Antipathidae | | | | | | | | | |
| *Antipathes* msp 1 | | | | 8.49 | | | | | |
| *Antipathes* msp 2 | | | | 0.11 | | | | | |
| *Stichopathes* msp 1 | | | | 9.35 | | | | | |
| Antipathidae indet. | | | | 0.54 | | | | | |
| Schizopathidae | | | | | | | | | |
| *Abyssopathes* cf. *lyra* | | | | | 0.50 | 0.56 | 0.58 | | |
| *Bathypathes* cf. *alternata* | | | | | | 0.14 | 0.19 | | 0.07 |
| *Bathypates* cf. *alternata* msp 1 | | | 0.08 | | | | | | |
| *Bathypathes* cf. *alternata* msp 2 | | 0.12 | | | | | | | |
| *Bathypathes* sp. | | | | | 0.19 | 0.14 | | | |
| *Bathypathes* msp 1 | | | 0.08 | | | | | | |
| cf. *Parantipathes* msp 1 | | | 0.11 | | | | | | |
| *Umbellapathes* aff. *bipinnata* | | 0.19 | 0.08 | | | | | | |
| *Umbellapathes* aff. *helioanthes* | | 0.58 | | | | | | | |
| Antipatharia indet. | 0.07 | 0.08 | 0.08 | 0.65 | 0.25 | 0.28 | 0.19 | | |
| Corallimorpharia/Corallimorphidae | | | | | | | | | |
| *Corallimorphus* msp 1 | | 0.04 | 0.00 | | | | | | |

| | | | | | | | |
|---|---|---|---|---|---|---|---|
| *Corallimorphus* msp 2 | | 0.46 | 0.08 | | 0.25 | 0.28 | 0.19 |
| Corallimorpharia msp 3 | | 0.04 | | | | | |
| Corallimorpharia msp 4 | | | 0.08 | | | | |
| Corallimorpharia msp A | | | | | 0.06 | | 0.19 |
| Corallimorpharia msp B | | | | | 0.06 | | |
| Scleractinia | | | | | | | |
| Scleractinia msp 1 | 0.14 | | | 7.85 | | | |
| Zoantharia | | | | | | | |
| Zoantharia msp 2 | | | | 0.11 | | | |
| Zoantharia indet. | | 0.46 | | 0.22 | | | |
| **Octocorralia** | | | | | | | |
| Alcyonacea | | | | | | | |
| Alcyoniidae | | | | | | | |
| *Anthomastus* msp 1 | 0.20 | | | | | | |
| *Anthomastus* msp 2 | 0.00 | 0.15 | | 0.11 | | | |
| Coralliidae | | | | | | | |
| *Corallium* sp. nov. | | | | 0.11 | | | |
| Chrysogorgiidae | | | | | | | |
| *Chrysogorgia* cf. *pinnata* | | | 0.08 | | | | |
| Isididae | | | | | | | |
| *Bathygorgia* aff. *abyssicola* 1 | | | | | | 0.14 | 0.19 |
| *Bathygorgia* aff. *profunda* 1 | | 0.15 | 0.08 | | | | |
| *Bathygorgia* aff. *profunda* 2 | | | 0.08 | | | | |
| *Keratoisis* aff. *flexibilis* msp 2 | | | 0.08 | | | | |
| Isididae msp 1 | | 0.04 | | | | | |
| Isididae indet. | 0.14 | | 0.76 | 0.11 | 0.13 | 5.63 | 2.31 |
| Taiaroidea | | | | | | | |
| Taiaroidae msp 1 | | | | | | | 0.19 |
| Primnoidae | | | | | | | |

| Taxon | 1 | 2 | 3 | 4 | 5 | 6 | 7 | 8 | 9 |
|---|---|---|---|---|---|---|---|---|---|
| *Abyssoprimnoa* cf. *gemina* | | | | | | 0.70 | 0.58 | | |
| *Callozostron* cf. *bayeri* | 0.07 | 0.54 | | | 0.06 | | | | |
| *Calyptrophora* cf. *persephone* | | | | | 0.06 | | | | |
| *Narella* msp 1 | | 0.08 | | 0.11 | | | | | |
| Primnoidea indet. | 0.61 | | 0.17 | | 2.70 | 3.38 | 1.54 | | |
| Alcyonacea msp 1 | | | | | 0.13 | | | 0.11 | 0.13 |
| Alcyonacea indet. | | 0.15 | 1.44 | 2.69 | 8.93 | 6.62 | 4.04 | | 0.07 |
| Pennatulacea | | | | | | | | | |
| Umbellulidae | | | | | | | | | |
| *Umbellula* msp 1_White | | | | | | | | | 0.13 |
| *Umbellula* msp 1_orange | | 0.31 | | 0.11 | | | | | |
| *Umbellula* msp 2 | | 0.08 | | | | | | | |
| Umbellulidae indet. | | 0.15 | | | | | | | |
| Protoptilidae | | | | 0.11 | | | | | |
| *Protoptilum* msp 1 | | 0.04 | | 0.22 | | | | | |
| Pennatulacea msp 2 | | 0.04 | | | | | | | |
| Pennatulacea msp 5 | | 0.23 | | | | | | | |
| Pennatulacea msp 6 | | 0.12 | | | | | | | |
| Pennatulacea msp 7 | | 0.38 | | | | | | | |
| Pennatulacea msp 8 | | 0.08 | | | | | | | |
| Pennatulacea indet | 0.14 | 2.08 | | 0.11 | | 0.14 | | | |
| Octocorallia msp 1 | | | | 0.22 | | | | | |
| Octocorallia msp 2 | | | | | | | | | |
| Anthozoa indet. | 0.14 | 0.12 | 0.51 | 0.65 | 0.13 | 0.14 | 0.19 | | |
| *Hydrozoa* | | | | | | | | | |
| *Branchiocerianthus* msp | | 0.08 | | | | | | | |
| Hydrozoa indet. | | 0.08 | 0.08 | 0.22 | | 0.14 | | | |
| **Crustacea*** | | | | | | | | | |

| | | | | | | | | | |
|---|---|---|---|---|---|---|---|---|---|
| ***Decapoda*** | | | | | | | | | |
| Caridea | 3.47 | 2.54 | 3.22 | 0.22 | 0.19 | | 1.15 | 0.11 | 0.20 |
| Decapoda msp 3 | | 0.08 | | | | | | | |
| Decapoda msp 4 | 0.07 | | | | | | | | |
| Decapoda/Aristeidae | 0.07 | 0.08 | | 0.54 | 0.06 | 0.56 | 0.58 | 0.11 | 0.07 |
| Decapoda msp 1 | | | | | | | | 0.06 | |
| Galatheidae | | | | | | | | | |
| Galatheidae small red msp | 2.79 | 0.54 | 0.17 | 0.43 | 0.06 | | | | |
| Galatheidae small white msp | 0.07 | 0.12 | | | | | | | |
| Munidopsis spp. | 0.82 | 0.35 | 0.51 | | | 0.42 | | | 0.20 |
| Galatheidae indet. | 0.14 | 0.15 | 0.17 | 0.11 | | | | | |
| Parapaguridae | | | | | | | | | |
| Parapaguridae msp 1/*Probeebei* sp. | 0.54 | 0.23 | | | | | | | |
| ***Peracarida*** | | | | | | | | | |
| Amphipoda | | | 0.08 | | 0.06 | 0.28 | | 0.06 | 0.27 |
| Podoceridae msp 1 | | | | | | | | 0.06 | |
| Amphipoda msp 1 | | 0.08 | 0.17 | | | | | | |
| Isopoda | | | | | | | | | |
| Munnopsidae msp 1 | | | | | 0.57 | 0.42 | 0.19 | 0.17 | 0.54 |
| Decapoda indet. | | 0.12 | 0.68 | | | | | | |
| Crustacea indet. | 0.07 | 0.31 | 0.51 | | | | | | 0.13 |
| | | | | | | | | | |
| **Echinodermata** | | | | | | | | | |
| ***Asteroidea*** | | | | | | | | | |
| Brisingida | | | | | | | | | |
| Brisingida msp 1 (6 arms - orange) | | 0.15 | 0.51 | | 0.25 | | | | |
| Brisingida msp 1 (8 arms - orange) | 0.14 | 0.38 | 0.25 | | | | | | 0.07 |
| Brisingida msp 3 (6 arms - white) | | 0.38 | 0.93 | 0.22 | 0.19 | 0.42 | 0.19 | | 0.13 |
| Brisingida msp 4 (9-10 arms) | 0.14 | 0.38 | | | | | | | |

| | C1 | C2 | C3 | C4 | C5 | C6 | C7 | C8 | C9 |
|---|---|---|---|---|---|---|---|---|---|
| Brisingida indet. | 0.27 | 0.08 | | 0.22 | | | | | |
| Paxillosida | | | | | | | | | |
| *Solaster* msp | | 0.04 | | | | | | | |
| Paxillosida cf AST_009/AST_007 | | 0.50 | 0.42 | | | | | | |
| Paxillosida msp 1 | 0.07 | | | 0.11 | | | | | |
| Paxillosida msp 2a | | 0.04 | | | | | | | |
| Paxillosida msp 2b | | | 0.08 | | | | | | |
| Paxillosida msp 3 | | 0.08 | 0.17 | | | | | | |
| Paxillosida msp 4 | | 0.08 | | | | | | | |
| Paxillosida msp 1 | | | | | | | | 0.06 | |
| Paxillosida indet. | | 0.65 | | | | | | | |
| Velatida | | | | | | | | | |
| Pterasteridae | | | | | | | | | |
| *Hymenaster* msp 2 | 0.07 | | | | | | | | |
| *Pteraster* msp | 0.20 | | | | | | | | |
| Velatida cf. AST_014 | 0.14 | 0.19 | | | | | | | |
| Velatida msp 2 | | | | | | | 0.19 | | |
| Velatida msp 3 | | | | | | | | | 0.07 |
| Asteroidea indet. | 0.48 | 0.42 | 1.10 | 0.11 | 0.19 | | | | |
| ***Crinoidea*** | | | | | | | | | |
| Comatulida | | | | | | | | | |
| Bourgueticrinina msp 1 | | | | | 0.31 | | | | 0.13 |
| Comatulida msp 1 | 1.97 | 1.54 | | | | | 0.19 | | |
| Comatulida msp 2 | | | | | | | | | 0.13 |
| Hyocrinida | | | | | | | | | |
| Hyocrinidae small msp | | | | | 0.38 | 0.28 | | | |
| Hyocrinidae msp 1 | | 0.19 | 0.08 | 0.00 | | | | | |
| Crinoidea red msp | 0.20 | 1.62 | | 0.43 | | | | | |
| Crinoidea golden msp | 0.14 | 0.38 | | | | | | | |

| | C1 | C2 | C3 | C4 | C5 | C6 | C7 | C8 | C9 |
|---|---|---|---|---|---|---|---|---|---|
| Crinoidea msp 1 | | | | | | | | | 0.07 |
| Crinoidea indet. | 0.07 | 0.46 | 0.08 | 0.11 | | 0.14 | | | |
| ***Echinoidea*** | | | | | | | | | |
| Aspidodiadematidae | | | | | | | | | |
| Aspidodiadematidae msp 1 | | | | | 3.96 | 2.68 | 2.31 | | |
| Aspidodiadematidae msp 2 | | 0.19 | | | | 0.14 | | | |
| Aspidodiadematidae soft msp | | | | 0.11 | | | | | |
| Aspidodiadematidae spiny msp | 0.14 | | | 0.75 | | | | | |
| Urechinidae | | | | | | | | | |
| Urechinidae msp 1_Nodules | | | | | | | | | 0.07 |
| Urechinidae msp 3 | 0.20 | 0.04 | 0.93 | | | | | | |
| Urechinidae msp 2_Nodules | | | | | | | | | 0.07 |
| Urechinidae msp 3_Nodules | | | | | 0.06 | | | | |
| Urechinidae msp 4_Nodules | | | | | | | | 0.06 | 0.13 |
| Urechinidae msp 1 | 0.20 | 0.73 | 0.25 | | | | | | |
| Urechinidae msp 2 | 0.20 | 0.04 | | | | | | | |
| Urechinidae msp 4 | 0.48 | 1.38 | 0.42 | | | | | | |
| Urechinidae msp 5 | 0.07 | | | | | | | | |
| Urechinidae msp 6 | 0.07 | | | | | | | | |
| Urechinidae msp 7 | 0.07 | | | | | | | | |
| Urechinidae indet. | 0.14 | 0.12 | 0.08 | | | | | | |
| Echinoidea indet. | 0.07 | | | | | | | | |
| ***Holothuroidea*** | | | | | | | | | |
| Elasipodida | | | | | | | | | |
| Elpidiidae | | | | | | | | | |
| Elpidiidae double-velum msp | | | | | | 0.19 | | | |
| Elpidiidae msp 1 | | | | | | | | 0.06 | 0.07 |
| *Amperima* msp | 0.14 | | | | | | | | |
| *Amperima* msp 1 | | | | | 0.06 | | | | |

| | C1 | C2 | C3 | C4 | C5 | C6 | C7 | C8 | C9 |
|---|---|---|---|---|---|---|---|---|---|
| *Peniagone* "palmata" msp | | | | | | 0.14 | 0.38 | | |
| *Peniagone* "tulip" msp | | | | | | | 0.19 | | |
| *Peniagone* cf. *leander* | | | | | | 0.14 | 0.19 | | |
| *Peniagone* msp | 0.14 | 0.08 | | | | | | | |
| *Peniagone* purple msp | | | | | | | | | 0.07 |
| *Peniagone* white/transparent msp | | | | | 0.06 | | | 0.06 | 0.07 |
| *Peniagone* indet. | | | | | 0.13 | | | | |
| Laetmogonidae | | | | | | | | | |
| Laetmogonidae msp 1 | 0.27 | 0.46 | | | | | | | |
| Laetmogonidae msp 2 | 0.20 | | | | | | | | |
| Laetmogonidae msp 3 | | | | | | | 0.19 | | 0.07 |
| Pelagothuriidae | | | | | | | | | |
| *Enypniastes* sp. | | | | | | | | | 0.07 |
| Psychropotidae | | | | | | | | | |
| *Benthodytes* cf. *incertae* purple msp | | 0.15 | 0.08 | | | | | | |
| *Benthodytes* cf. *incertae* red msp | | 0.42 | | | | | | | |
| *Benthodytes* msp | | 0.19 | | | | | | | |
| *Benthodytes* msp 1 | | | | | | | | | 0.07 |
| *Benthodytes* pink msp | | | | 0.11 | | | | | |
| *Benthodytes* purple msp | | | 0.08 | | | | | | |
| *Benthodytes* red msp | | 0.04 | 0.08 | | | | | | |
| *Psychropotes* cf. *semperiana* | | | | | | | | 0.06 | |
| *Psychropotes longicauda* | | | | | | | 0.38 | | |
| *Psychropotes* msp 3 | | | | | 0.06 | | 0.19 | | |
| *Psychropotes verrucosa* | | | | | 0.25 | 0.14 | | | |
| Psychropotidae msp 1_Nodules | | | | | 0.06 | 0.14 | 0.19 | | |
| Psychropotidae msp 1 | | 0.35 | 0.08 | | | | | | |
| Psychropotidae msp 2_Nodules | | | | | | 0.42 | | | |
| Psychropotidae msp 2 | | 0.04 | | | | | | | |

| Taxon | | | | | | | | |
|---|---|---|---|---|---|---|---|---|
| Psychropotidae msp 3 | | | | | 0.13 | 0.14 | | |
| Psychropotidae msp 4 | | | | | | 0.14 | | |
| Psychropotidae red msp | 0.14 | | | | | | | |
| Psychropotidae indet. | 1.22 | 0.42 | | 0.11 | | | | |
| Holothuriida | | | | | | | | |
| Mesothuriidae | | | | | | | | |
| *Mesothuria* msp | 0.07 | 0.12 | | | | | | |
| Synallactida | | | | | | | | |
| Deimatidae | | | | | | | | |
| *Deima* msp. | | 0.04 | | | | | | |
| Deimatidae - irregular papillae length msp | | 0.27 | 0.08 | | | | | |
| *Oneirophanta* msp | 0.07 | | 0.17 | | | | | |
| Deimatidae indet. | | 0.04 | 0.08 | | | | | |
| Synallactidae | | | | | | | | |
| *Benthothuria* msp | | | | 0.43 | | | | |
| *Paelopatides* "orange" msp | 0.07 | 0.04 | | | | | | |
| *Synallactes* msp 1 (Synallactidae purple msp) | 0.07 | | | | | | | |
| *Synallactes* msp 2 | | 0.04 | | | | | | |
| *Synallactes* msp 2 pink | | | | | 0.13 | 0.56 | 0.19 | |
| *Synallactes* msp 2 pink (smooth) | 0.20 | 0.08 | | | | 0.70 | | |
| *Synallactes* sandy-coloured msp | 0.14 | | | | | | | |
| *Synallactes* white msp | 0.14 | | | | 2.33 | 0.42 | 0.96 | 0.07 |
| Synallactidae indet. | 0.27 | | | | | | | |
| Persiculida | | | | | | | | |
| Molpadiodemidae | | | | | | | | |
| Molpadiodemas msp | | 0.12 | | | | | | |
| Pseudostichopodidae | | | | | | | | |
| Pseudostichopus msp | | | | | | 0.14 | | |
| Molpadiodemas/Mesothuria | | | | | 0.19 | 0.28 | 0.19 | 0.13 |

| Taxon | | | | | | | | | |
|---|---|---|---|---|---|---|---|---|---|
| Holothuroidea indet. | 1.29 | 0.73 | 0.25 | 0.22 | 0.19 | 0.14 | 0.38 | | |
| *Ophiuroidea* | | | | | | | | | |
| Ophiuroidea msp 1 | | | | | 0.06 | 0.14 | 0.19 | | |
| Ophiuroidea msp 3 | | | | | | 0.28 | | | |
| Ophiuroidea msp 5 | 0.14 | 1.92 | 3.39 | | | | | 0.06 | 0.20 |
| Ophiuroidea msp 6 | | 0.15 | 0.08 | | 1.07 | 2.96 | 2.12 | 0.34 | 0.87 |
| Ophiuroidea msp 4 | 0.27 | 1.04 | | | 0.38 | | | | |
| Ophiuroidea msp7 | | 0.04 | | | | | | | |
| Ophiuroidea indet. | | 0.12 | 0.25 | 0.43 | 18.93 | 15.07 | 23.65 | | 0.27 |
| | | | | | | | | | |
| **Enteropneusta** | | | | | | | | | |
| Enteropneusta msp 1 cf. *Yoda* | | 0.50 | | | | | | | |
| Enteropneusta msp 2 cf. *Saxipendum* msp. | 0.54 | | | | | | | | |
| | | | | | | | | | |
| **Mollusca** | | | | | | | | | |
| ***Gastropoda*** | | | | | | | | | |
| Limpet | | | 0.08 | | | | | | |
| Gastropoda msp 1 | | | 0.17 | | | | | | |
| Polyplacophora | 0.27 | | | 0.22 | | | | | 0.13 |
| Gastropoda indet. | | | | | | 0.14 | | | |
| ***Cephalopoda*** | | | | | | | | | |
| Octopoda msp 1 | 0.07 | | | | | | | | |
| | | | | | | | | | |
| **Pisces*** | 2.52 | 1.38 | 0.51 | 0.32 | 1.57 | 0.42 | 1.54 | 0.34 | 0.34 |
| | | | | | | | | | |
| **Porifera** | | | | | | | | | |
| ***Demospongiae*** | | | | | | | | | |
| Cladorhizidae | | | | | | | | | |
| Cladorhizidae msp 1 | | | 0.17 | | | | | | 0.13 |

| | 1 | 2 | 3 | 4 | 5 | 6 | 7 | 8 | 9 |
|---|---|---|---|---|---|---|---|---|---|
| Cladorhizidae msp 1(soft) | | | | | | | | | 0.07 |
| Cladorhizidae msp 2 | | | | | | | | 0.06 | 0.07 |
| Cladorhizidae msp 3 | | | | | | | | | 0.07 |
| Cladorhizidae msp 4 | | | | | 0.06 | | | 0.11 | 0.27 |
| Cladorhizidae msp 5 | | | | | | 0.14 | | | |
| Cladorhizidae msp 6 | | | | | | 0.14 | | | |
| Cladorhizidae indet | | | | | | | | 0.06 | 0.13 |
| ***Hexactellinida*** | | | | | | | | | |
| Euplectellidae | | | | | | | | | |
| *Bathydorus spinosus* | 0.07 | | | | | | | | |
| *Bolosoma* sp. | | | | 0.11 | | | | | |
| *Corbitella discasterosa* | 0.07 | | | | | | | 0.11 | |
| *Docosaccus maculatus* | | | | | | 0.14 | | 0.06 | |
| *Docosaccus nidulus* | | | | | | 0.14 | | | |
| *Holascus* spp | | | | | 0.63 | 0.28 | 0.19 | 0.06 | 0.07 |
| *Hyalostylus schulzei* | | | | | | | | 0.06 | |
| *Hyalostylus* sp. | | 0.08 | 1.02 | 0.11 | | | | 0.06 | 0.07 |
| *Sacocalyx pedunculatus* | | | | 0.32 | | | | | |
| *Sacocalyx* sp. | 0.27 | 0.12 | 0.17 | | | | | | |
| Euretidae | | | | | | | | | |
| *Bathyxiphus subtilis* | | | | | | | | 0.06 | |
| *Chonelasma bispinula* | | | | | | | 0.19 | | |
| *Chonelasma choanoides* | 0.07 | | | | | | | | |
| *Chonelasma* sp. | | | 0.08 | | | | | | |
| Hyalonematidae | | | | | | | | | |
| *Hyalonema* spp. | | 0.08 | 0.68 | 0.11 | 0.38 | 0.70 | 0.77 | 0.17 | 0.47 |
| Rosselidae | | | | | | | | | |
| *Caulophacus* sp. | | 0.31 | 0.51 | 0.11 | 0.57 | 0.14 | 0.19 | | |
| *Crateromorpha* sp. | | 0.08 | | 0.11 | | | | | |

| | | | | | | | | | |
|---|---|---|---|---|---|---|---|---|---|
| Rossellidae gen. sp. | 0.27 | 0.04 | 0.17 | | | | | | |
| Pheronematidae | | | | | | | | | |
| Poliopogon sp. | | | | 0.11 | | | | | |
| Hexactellinida/foliose sponge msp | 0.07 | 0.12 | 0.08 | 0.32 | | | | | |
| Hexactellinida - Stalked | | | | | 0.88 | 1.13 | 1.73 | | |
| Hexactinellida black msp | | 0.04 | | | | | | | |
| Hexactellinida indet. | 1.50 | 1.00 | 3.56 | 0.65 | 3.27 | 2.39 | 5.19 | 0.89 | 0.74 |
| | | | | | | | | | |
| **Pycnogonida** | 0.14 | 0.00 | 0.08 | 0.00 | | | | | 0.07 |
| | | | | | | | | | |
| **Tunicata** | | | | | | | | | |
| Octacnemidae | | | | | | | | | |
| *Megalodicopia* msp. 1 | 0.14 | 0.04 | 0.08 | | | | | | |
| *Megalodicopia* msp. 2 | | | | | | | | | 0.07 |
| *Dicopia* msp. | 0.27 | | | | | | | | |
| Pyuridae | | | | | | | | | |
| *Culeolus* msp. | | | | | | | | | 0.07 |
| Tunicata indet. | 0.14 | 0.04 | 0.08 | 0.11 | | | | | |
| | | | | | | | | | |
| *Paleodictyon nodosum* | | | | | | | | 0.06 | |

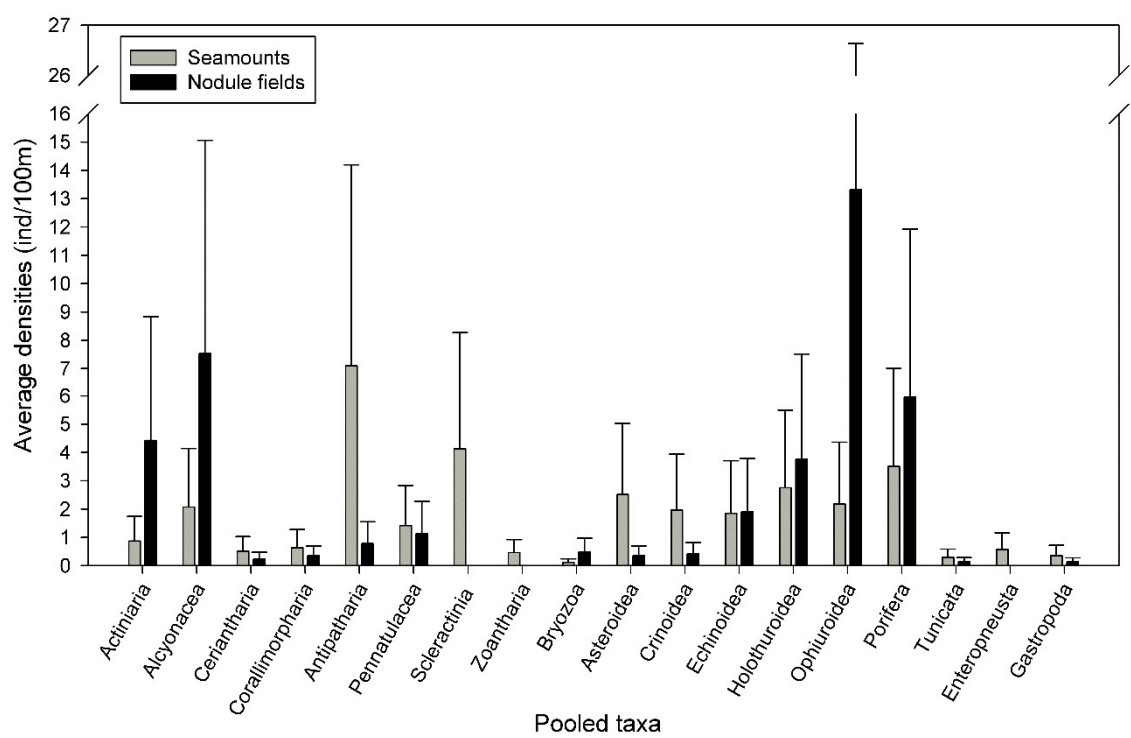

Fig. A1. Average densities at higher taxa level per ecosystem and standard deviation.