# Peer review of "Are seamounts refuge areas for fauna from polymetallic"

_Biogeosciences, 2019_

## Referee Comment (RC1) · Anonymous Referee #1 · 2 Oct 2019

General evaluation

The paper addresses the question whether seamounts may act as refuges for fauna that will potentially be affected by large-scale mining for polymetallic nodules. It is based on the assumption that the occurrence of hard substrate at both habitat types may result in similar megafaunal communities. Using ROV video transects, seamounts and nodule fields in the CCZ area were compared with respect to composition and abundance of megafauna. The authors conclude that the observed substantial differences between seamount and nodule field communities make it highly unlikely that seamounts can compensate for the impacts of deep seabed mining.

The recent efforts to promote deep-sea mining for minerals, which, in the case of nodule mining, will involve the destruction of large seabed areas including its faunal com-

munities, make the paper highly relevant in the light of the need to preserve biodiversity in the deep sea. Although the paper is purely descriptive, it could add to the knowledge of biodiversity in the CCZ and particularly at the seamounts in the area, which had not been sampled before, and make a valuable contribution to the question whether deep seabed mining can be responsibly managed. However, the paper has several weaknesses, which have to be addressed before publication. My main concern is that no environmental data are presented. In additon to missing hydrographical data, there is no detailed description of the sampling sites, particularly the seamounts, such as size, summit and base depth, inclination of slopes/general bathymetry, or current field. More important, no information is given on habitat types encountered along the transects. It is well known that substrate can vary considerably at short distances at seamounts, and of course megafaunal communities are strongly associated with substrate type. This is briefly mentioned in the discussion, but I do not understand why this information is not provided and analysed in the results. It should easily be available from the video footage.

There are also some methodological issues. The basic problem, as also admitted by the authors in the discussion, is that only a very limited number of rather short transects without replications are available and that transects at the seamounts and at the nodule fields were taken at different depths; in the case of Mann Borgese Smt the depth sampled was nearly 3000 m less than on the corresponding nodule field, and hence the data are hardly comparable. Although the depth difference at the other sites was much smaller, it may also limit the comparability of the data. This is mentioned in the discussion, but the consequences should be elaborated in more detail, and it makes the conclusion that "seamounts appear inadequate as refuge areas to help maintain nodule biodiversity" disputable. I am also not convinced that the quantification of the samples is correct. In section 2.1, the authors state that the altitude of the ROV was "kept constant whenever possible". Apart from not providing the information at which target altitude the ROV was kept, and whether this was the same at all transects, the authors inform in section 2.2 that, due to varying altitude as well as pan and tilt

of the camera, "surface coverage" could not be used for standardisation and instead just transit length was used. However, since the field of view and thus the number of visible objects per unit transit section depend on the altitude and angle of the camera, the standardisation to 100 m transit sections, without taking into account the varying field of view, could strongly bias the results. Generally, the methods section has to be improved with much more detail.

Further, it is not clear how the investigations made in this study relate to those by Vanreusel et al (2016) who also presented results for epifaunal communities in the CCZ, comparing APEI, BGR, GSR and others. Obviously the sampling was done on the same cruise using the same gear. Were the same nodule field transects analysed? If yes, this has to be justified, the additional value of this study as compared to Vanreusel et al (2016) has to be demonstrated (apart from the additional seamount transects) and any overlap and differences in the analysis indicated. If not, a thorough comparison between the results of both studies is necessary.

There are numerous smaller issues throughout the text, including discrepancies between abundances given in the text and in Tab. A1. Details are given below.

Because of the scarcity of information on faunal assemblages in the deep sea in general and particularly at seamounts in the area of the CCZ I suggest to consider the paper for publication despite its weaknesses, but only after major revision, taking into account the comments above and the suggestions for improvements given below.

Specific comments

- Use consequently "Ophiuroida, Asteroida" etc. instead of "ophiuroids, asteroids" etc.

- Abundances given in the text are not always consistent with those presented in Tab. A1. I did not check all entries, but two examples caught my eye: A total abundance of 89.2 ind/100 m is given for ROV10 in Tab. 1 and in the text, but summing up all observations in Tab. A1 results in ca. 67 ind/100 m. Another example: For Porifera,

numbers given in the text match those in Tab. A1 for Rüppel and Senckenberg, but those presented for Heip and Mann Borgese are much lower than in the table (3 vs. 7.5 and 0.68 vs. 1.9, respectively). This has to be checked and resolved.

- Line 42: Insert common definition for "seamount" and citation

- 73: When did the sampling take place?

- 96: What is the difference between "exploration and opportunistic sampling"? More detail is needed.

- 98: What does "whenever possible" mean? 90 % of the transects? What was the target altitude of the ROV, and was it the same at all transects? How did panning and tilting affect the field of view? (see also general comments).

- 94-99: Generally, much more information on the sampling mode is necessary, including sampling strategy (e.g., straight line, deviations for interesting objects etc.), ROV speed, target altitude, field of view etc.

- 104: What is "ID's"? I guess it should be IDs, but "ID" is not defined in the text. Same in line 111.

- 114: Is there a reason that specimens collected were obviously not used for proper identification?

- 118: Which statistical testing? Did the authors use tests other than nMDS? If yes, they have to be described here in detail

- 127: Here and throughout the text: two significant digits are sufficient, for example 7.6 instead of 7.59 or 89 vs. 89.23. The two decimals pretend a non-existing precision of the data.

- 151/152: Aren't Acrocirridae polychaetes as well? (". . . Acrocirridae were observed. . . as well."). Do you probably mean they were observed in high densities in some of the transects?

- 189/190: This belongs into the discussion.

- 193: Insert ", respectively" after "Table A1"

- 195: "less linear" - how was this assessed? I cannot see any linear or non-linear relations in Fig. 4d, nor can I see any curves crossing.

- 198: What does "small sample size" mean? I think the sample size in this study was always small.

- 205: Should read "least overlap". Explain similarity between these findings and the results from the seamount: For both habitat types, the samples at APEI3 had least overlap with the other sites.

- 220ff: According to Fig. 3 (not Fig. 8!), the majority of ophiuroids on the nodule fields were unidentified

- 244/245: This is not clear. Variation "along the video transects" was obviously not analysed and cannot be seen in Fig. 5. Probably the authors mean "between transects"?

- 251: Kendall's coefficient is not mentioned in methods section. See comment above.

- 255-258: This makes no sense. If sampling depth differs between seamounts and NF, and nMDS distinguishes between seamount and NF groups, then the grouping must correspond also to depth sampled. Omit this paragraph (and Fig. 7b) and state in the discussion that differences between seamounts and NF could be a result of different depths sampled.

- 262: This is not quite clear. Rephrase: ". . . at different locations and additionally, for the seamounts, different depth ranges." Possible differences in substrate etc. should be mentioned here.

- 269: Rephrase: ". . . since (mega)faunal communities could be very different even between adjacent seamounts . . ."

- 270: Which parameters? Name examples for depth-dependent parameters which drive faunal composition

- 278: Why would "elevated topography (peaks)" favour Porifera and Anthipataria? Name possible mechanism(s). By the way: Seamounts are per definitionem elevated topography.

- 288-295: Do the authors mean that faunal density is negatively correlated with nodule coverage? This is in contrast to Vanreusel et al 2016, who found higher abundances at higher nodule coverage. So obviously in this study, the driver for the differences in faunal density was not nodule coverage, but probably organic input.

- 296: Grammar: neither - nor

- 319: Clearly distinguish between own data and data from literature by rephrasing, e.g. "Vanreusel et al. (2016) found that ophiuroids…."

- 322 : The available data cannot show a gradient, therefore it should read: ">50% less at seamounts compared to nodule fields"

- 331: "… studied here."

- 338: How can an uneven distribution (of holothuroids) affect composition?

- 350: This is an isolated statement here - what does it imply?

- 352: And what about nodule-covered areas - did they host these taxa in Vanreusel's or this study? This paragraph is a bit confusing and should be re-sorted, also clearly distinguishing between own results and those of others.

- 361: "communities"

- 362: "… were more abundant…" - compared to what?

- 370: "…they are known to …" Citation?

372: Does the reference (Baco 2007) apply to both statements? I suggest to rephrase,

e.g. "The exception. . .common on seamounts, as also reported in other studies (e.g., Baco. 2007)." Baco 2007 is not in the reference list!

- 373: Insert: ". . . Enteropneusta which in this study were found only on seamounts, were. . ."

Figures

- Fig. 1: What does the "A" in the upper left corner mean?

- Fig. 2: I suggest to add morphospecies "names" (as given in Fig. 3) to the examples.

- Fig. 3: This figure should be simplified. Most of the morphospecies were observed in very low numbers, and in these cases differences between NF and seamounts are difficult to see in the figure and rather not relevant. I suggest to include in this figure only morphospecies and higher taxa with a substantial mean abundance (e.g., >0.5 ind./100 m per habitat type); other morphospecies could be summarised or omitted. By contrast, Table A1 should be extended and present the results for all morphospecies, not only higher taxa (see below).

- Fig. 4: Axis labels are incomplete (units are missing). What does "exact" on the y-axis in panels a and c mean? And what is sample size (units?) in panels b and d? I guess that not sample size was used for the rarefaction curves, but accumulated number of observations. Caption is incomplete: What do the shaded areas in panels a and c and horizontal and vertical lines in panels b and D mean?

- Fig. 5: What does "values are relative" mean? - percent (of what?)? This has to be explained.

- Fig. 6: y-axis incomplete, should include quantity and unit.

- Fig. 7: Omit panel b).

Tables

- Tab. 1: Be consistent with units: here, #obs/100 m is given, whereas throughout the text and in figures and in Tab. A1 the unit for density is ind/100 m.

- Tab. A1: See also comment to Fig. 3. I suggest to list data for all distinguished morphospecies here and sums for higher taxa. It is irritating that densities for higher taxa (e.g., Holothuroidea) are given, but they do not include the identified morphospecies within that taxon. Not identified taxa should be clearly indicated, e.g. Holothuroidea indet., and they should sum up with the distinguished morphospecies to total Holothuroidea, etc. I also suggest to include absolute number of observations in addition to densities in this table. This would facilitate the evaluation whether, e.g., differences are based on a substantial number of individuals, or rely on just one or two specimens. The order of taxa in the table is not clear.

---

## Referee Comment (RC2) · Anonymous Referee #2 · 10 Oct 2019

The paper by Cuvelier et al is an interesting analysis of ROV observations in the CCZ. They test the seamount refugia hypothesis to the extent possible with sparse semi-quantitative transect data. The results are somewhat equivocal but the authors provide many caveat and qualifications in the text. Overall their main conclusion that seamounts are unlikely refuges for taxa living on minable nodule fields is likely accurate because its quite clear that faunal overlap is low between these habitats. There are a few comments below that could be addressed to improve the paper.

Major comments 1) The appendix fig1 is a very important figure to place all the observations into context. I would move it from an appendix to a regular figure for the paper or as new panels in existing Fig. 1. 2) The data presentation used to compare nodule transects to seamount transects should be refined. Right now figure 3 portrays averages of densities for fine taxa which, based on the finding of very low overlap between transects, means little - averaging a large number for one transect with low or zero numbers in other transects. Rather the data should be presented at broader taxonomic categories (as in Appendix table 1) with average (and standard deviation) densities and # of morphospecies. This would also follow the language in the results section better. Data on each fine morphospecies (level of taxonomy in Fig3) could be presented an an appendix and by transect. 3) Once data is pooled at higher taxa levels, statistical comparisons could be drawn to compare the average # morphospecies and average density between nodule and seamount transects. 4) The authors conclude that the ratio of hard/soft habitat may explain some of the faunal trends they observe. Can't this ratio be determined from the transects? If possible add this metric to help explain faunal communities along or between the transects.

Specific comments line 82 - Explain why the north or northwestern flanks of the seamounts were chosen for the transects. line - 97-99 - Provide the range of altitude, speed that were kept constant. line 208-209 - this statement appears true when examining the higher taxa pooled data in appendix table 1. however in the cited fig 3, its hard to actually make this comparison because averages at finer taxonomic categories are highly variable due to lack of fine taxa overlap between transects. line 267 - Start the sentence with, "Amongst the seamount transects,..." line 269 - The point of this sentence is not clear as it opposes the trend you find. Clarify. lines 296-306 - Nice to see a paragraph which lays out what future transecting should look like. The paragraph mentioned that wider depth ranges should be included and the data and literature certainly support that. Might it also be wise to have transects that move along countours so there are many replicate obsrevations at a given depth, instead of conducting uphill transects? Adding a sentence or two addressing across slope vs with slope transecting would be worthwhile. line 308 - The sentence should be slightly reworded based on the authros findings to "Seamounts were shown to share FEW fauna with surrounding habitats...." Line 316 - this topic sentence needs to be improved. Rather than simply reiterating the results section can this paragraph be rewritten and

a topic sentence created that summarizes the functional differences between taxa on seamounts vs nodules?. See literature by Rowden etal that look at functional variation of taxa on seamounts and neighboring areas. E.g. Rowden et al 2010 Marine Ecology Table 1 - Add "SM:" before Mann Borgese figure 3 - given that there is so little overlap in the morphospecies between each sampled transect Figure 3 is a bit hard to interpret. Error bars would help. Its great that the taxonomic diversity is presented but this might be better in the appendix. Instead, appendix table 1 which presents higher taxa and # morphospecies might be the better data to show in the main paper. Here densities at higher levels can be better compared. Averageing the abundances across the transects seems ill advised given the differences observed between each one. Figure 6 - It is not clear what data is being presented here. Are these only taxa present on both seamounts and nodules? Please clarify in the figure caption.

―――――――――――――――――――

---

## Author Comment (AC2) · 31 Oct 2019

We would like to thank the reviewer for their point of view and suggestions that contributed to the revision of the manuscript. We have added a supplement .pdf, with the same contents as listed below, but featuring our replies in blue and paragraphs altered or added in italic.

There are a few comments below that could be addressed to improve the paper. Major comments

R2: 1) The appendix fig1 is a very important figure to place all the observations into context. I would move it from an appendix to a regular figure for the paper or as new panels in existing Fig. 1.

A: We have incorporated it as an extra panel in Figure 1.

R2: 2) The data presentation used to compare nodule transects to seamount transects should be refined. Right now figure 3 portrays averages of densities for fine taxa which, based on the finding of very low overlap between transects, means little - averaging a large number for one transect with low or zero numbers in other transects. Rather the data should be presented at broader taxonomic categories (as in Appendix table 1) with average (and standard deviation) densities and # of morphospecies. This would also follow the language in the results section better. Data on each fine morphospecies (level of taxonomy in Fig3) could be presented an an appendix and by transect.

A: We decided to add morphospecies to Table A1 (also taking into account the comments made by the other reviewer), which thus give information per transect, help to elucidate Fig. 3 and add the desired level of taxonomy. Figure 3 was withheld because it was considered a key figure to show how different the presence/absence/abundance of the fauna varies between the two ecosystems, but the 3 parts were separated more clearly and different breaks at the X-axis, thus enhancing readability and interpretability. This information was added in the figure caption as well.

R2: 3) Once data is pooled at higher taxa levels, statistical comparisons could be drawn to compare the average # morphospecies and average density between nodule and seamount transects.

A: There were no significant differences for the densities (T-tests for samples with unequal variance) per taxon (taxa pooled and tested: Actiniaria, Alcyonacea, Ceriantharia, Corallimorpharia, Antipatharia, Pennatulacea, Scleractinia, Zoantharia, Bryozoa, Asteroidea, Crinoidea, Echinoidea, Holothuroidea, Ophiuroidea, Porifera, Tunicata, Enteropneusta, Gastropoda. The pooled data were visualised as a histogram with st. dev., added as an extra figure to this reply to the reviewer (Fig. R1), and could be added to the appendix. The number of morphospecies per higher pooled taxon group proved more difficult, since often we know there are >1 or >2 morphospecies, but not

the exact numbers. Using the minimum known number would be and under-estimation, which is why we chose not to test these pooled observations.

R2: 4) The authors conclude that the ratio of hard/soft habitat may explain some of the faunal trends they observe. Can't this ratio be determined from the transects? If possible add this metric to help explain faunal communities along or between the transects.

A: We have added information on the amount of hard substrata under the form of 3 categories: (1) Predominant soft substrata (<40% hard substrata), (2) mix or transition (40-60% hard substrata) and (3) predominant hard substrata (>60% hard substrata), annotated over 10m distance units. Very few significant relationships were revealed (only for Mann Borgese ROV15), though backscatter data is currently being analysed to model the geomorphology along the transects in more detail and help reveal more details of fauna/substratum relationships, but it is out of the scope of this article. The following paragraph was added in section 3.1 (also taking into account the other reviewer's suggestions): "About 57% of all sessile fauna was associated with predominantly hard substrata, followed by 31% on the mixed substrata. For the mobile taxa the pattern was less pronounced with 41 and 42% associated with predominantly hard and mixed soft/hard substrata respectively. The amount of predominantly hard and soft substrata were negatively correlated though, not significantly. This was due to the elevated amount of mixed hard/soft substrata featuring equal amounts 40-60%. Over all seamount transects pooled together, no taxa were significantly correlated with the amount of hard substrata, nor with soft substrata. When looking at the individual transects, no significant correlations were found between taxa and substrata for ROV02 or ROV04 or ROV09, most likely due to the equal distribution of the amount of hard/soft/mix substrata. In this perspective, ROV15 stood out, as it was dominated by predominantly hard substrata (56/%): For this transect, Pennatulacea were significantly negatively correlated with the amount of hard substrata and Zoantharia/Octocorralia were significantly and positively correlated with hard substrata, as were Ophiuroidea,

Asteroidea, Crinoidea and Mollusca."

Specific comments

R2: line 82 - Explain why the north or northwestern flanks of the seamounts were chosen for the transects.

A: These flanks were chosen based on the positioning of the vessel and the predominant surface current in order to avoid the umbilical of the ROV to drift/being transported towards the vessel. Predominant currents in the CCZ are South-East oriented, allowing for an ROV positioning "downstream" of the vessel's location while visiting the north-northwestern seamount flanks. We added this as follows: "The four seamount transects were characterised by different depth ranges and lengths and, due to the vessel's positioning and the predominant South-East surface currents, were all carried out on the north to north-western flanks of the seamounts (Table 1 and Fig. 1)."

R2: line - 97-99 - Provide the range of altitude, speed that were kept constant.

A: Target altitude was <2m above seafloor and travel speed ∼0.2m/s, though interrupted by sampling actions, instrument check-ups, exploration, object avoidance (in the case of the uphill seamount transects) etc. This was added in the body of text.

R2: line 208-209 - this statement appears true when examining the higher taxa pooled data in appendix table 1. however in the cited fig 3, its hard to actually make this comparison because averages at finer taxonomic categories are highly variable due to lack of fine taxa overlap between transects.

A: This is amended by adding the morphospecies information in Table A1.

R2: line 267 - Start the sentence with, "Amongst the seamount transects,..."

A: Ok

R2: line 269 - The point of this sentence is not clear as it opposes the trend you find. Clarify.

A: This sentence was to point out that it could rather be depth influencing their similarity than their adjacent location. This part was changed (also taking into account the comments from the other reviewer) as follows: "For seamounts, distance separating them might be a less determining factor than depth, since (mega)faunal communities can be very different even between adjacent seamounts (Schlacher et al., 2014; Boschen et al., 2015). Overall, parameters that vary with depth, such as temperature, oxygen concentration, substratum type, food availability, and pressure are considered major drivers of species composition on seamounts (Clark et al., 2010; McClain et al., 2010)."

R2: lines 296-306 - Nice to see a paragraph which lays out what future transecting should look like. The paragraph mentioned that wider depth ranges should be included and the data and literature certainly support that. Might it also be wise to have transects that move along countours so there are many replicate obsrevations at a given depth, instead of conducting uphill transects? Adding a sentence or two addressing across slope vs with slope transecting would be worthwhile.

A: This is a valid observation and we added the following sentence to this paragraph: "Alternatively, across slope transects, following depth contours, should be considered as these could provide observation replicates for a given depth"

R2: line 308 - The sentence should be slightly reworded based on the authros findings to "Seamounts were shown to share FEW fauna with surrounding habitats...."

A: Based on the literature, fauna from seamounts tend to occur in neighbouring habitats quite easily, but that is not the case here. We rephrased the sentence to clarify this. "In other areas, seamounts were shown to share fauna with surrounding habitats (Clark et al., 2010) and could thus potentially serve as source populations for neighbouring environments (McClain et al., 2009)."

R2: Line 316 - this topic sentence needs to be improved. Rather than simply reiterating the results section can this paragraph be rewritten and a topic sentence created that

summarizes the functional differences between taxa on seamounts vs nodules?. See literature by Rowden etal that look at functional variation of taxa on seamounts and neighboring areas. E.g. Rowden et al 2010 Marine Ecology

A: We added in topic sentences for several paragraphs and cleaned up the remainder of the body of text as to not repeat the results. The section currently reads as follows: "Overall, nodule fields showed higher faunal densities than seamounts. Such shifts in density patterns between nodule fields and seamounts were clearer in a number of taxa, where the variety of morphospecies and feeding strategy within each group was likely to be at play. One taxonomic group in which this was rather clear were the Echinodermata, which group Asteroidea (predators and Brisingid filter feeders), Crinoidea (filter feeders), Echinoidea (Deposit feeders), Holothuroidea (Deposit feeders) and Ophiuroidea (Omnivores). Ophiuroidea were most abundant on the nodule fields (ratio 7 to 1 when compared to seamounts). Asteroidea and Echinoidea (with exception of one very abundant morphospecies at the nodule fields) were both more abundant and diverse on the seamounts. Same ophiuroid morphospecies were present at seamounts and nodule fields but in very different abundances and they showed preference for different substrata, which also appeared to correspond to different lifestyles, feeding behaviour and corresponding dietary specialisations (Persons and Gage, 1984). Previously it was already demonstrated that Ophiuroidea did not show high levels of richness or endemism on seamounts (O'Hara, 2007). At nodule fields Ophiuroidea were often observed in association with xenophyophores (Amon et al., 2016, this study) and a similar observation was done at east Pacific seamounts off Mexico (Levin et al., 1986), though no such associations were observed on the seamounts studied here. Holothuroidea densities were thought to possibly decrease when less soft sediment was available since they feed mainly on the upper layers of the soft-bottom sediment (Bluhm and Gebruk, 1999). No significant link was established between holothuroid densities and the amount of hard substrata, but their community composition varied distinctly between nodule fields and seamounts with more families being observed at the latter. Additionally, at the seamounts, many holothurians were observed on top of

rocks, possibly reflecting different feeding strategies and explaining the observations of different morphospecies. Geographical variations, different bottom topography, differences in nodule coverages and sizes and/or an uneven distribution of holothurians on the sea floor were thought to play a role in holothuroid community composition (Bluhm and Gebruk, 1999). On the other hand, variability in deep-sea holothuroid abundance was proposed to depend primarily on depth and distance from continents (see Billet, 1991 for a review). Stalked organisms, such as Crinoidea (Echinodermata) and Hexactinellida (except for Amphidiscophora, Porifera) rely on hard substrata for their attachment and are considered being among the most vulnerable organisms when mining is concerned. Crinoidea were more abundant on seamounts, possibly because hard substrata were less limiting than in the nodule fields. Porifera densities (stalked and non-stalked) varied among all analysed transects, revealing no particular trends in abundance. However, the species composition of deep-sea glass sponge communities from seamounts and polymetallic nodule fields was distinctly different. Polymetallic nodule field communities were dominated by widely-distributed genera such as Caulophacus and Hyalonema, whereas seamount communities seemed to have a rather unique composition represented by genera like Saccocalyx. Corals were considered to be more abundant on seamounts than adjacent areas, due to their ability to feed on a variety of planktonic or detritus sources suspended in the water column, (Rowden et al., 2010). In this study, the Alcyonacea densities were lower at the seamounts than on the studied nodule transects. The Antipatharia were most abundant at the Mann Borgese seamount (APEI3) compared to all other transects, seamounts and nodule fields. The depth difference of more than 3000m between this particular seamount and the nodule fields could explain the abundance in Antipatharia which were shown to be more abundant at lower depths (Genin et al., 1986). The Antipatharia and Alcyonacea morphospecies of the seamounts did not occur on the nodule fields and vice versa, with exception of Callozostron cf. bayeri which was present at the nodule fields but in very low densities (1/10 of those observed at seamounts). Additional presence of Pennatulacea, which were virtually absent from the nodule field transects,

resulted in completely distinct coral communities for each ecosystem. Actiniaria were denominated the second most common group at CCZ nodule fields, after the xenophyophores (Kamenskaya et al., 2015) and, in our study, were also more abundant on nodule fields than on seamounts. Depending on the species and feeding strategy, the ratio hard/soft substrata and their preference for either one could play a role. Since morphospecies were distinct between seamounts and nodule fields, their role in the respective communities are likely to differ as well. Combinations of deposit feeding and predatory behaviour in Actiniaria have been observed, as well as burrowing activity, preference for attachment to hard substrata and exposure to currents (Durden et al., 2015a; Lampitt and Paterson, 1987; Riemann-Zürneck, 1998)." On a side note, functional traits of seamount and nodule field fauna are being investigated in a broader framework, extending beyond the feeding group and including life history, mobility etc.

R2: Table 1 - Add "SM:" before Mann Borgese

A: This was added to the table.

R2: figure 3 - given that there is so little overlap in the morphospecies between each sampled transect. Figure 3 is a bit hard to interpret. Error bars would help. Its great that the taxonomic diversity is presented but this might be better in the appendix.

A: We changed the Y-axis of the different parts of the figure 3 graph to make them more easily interpretable. See reply above. We have uploaded an extra figure linked to this reply to the reviewers that can be included in the appendix, pooling the densities into larger taxa.

R2: Instead, appendix table 1 which presents higher taxa and # morphospecies might be the better data to show in the main paper. Here densities at higher levels can be better compared. Averageing the abundances across the transects seems ill advised given the differences observed between each one.

A: We have added the morphospecies' densities to the appendix table, which also

clarifies the taxonomy and figure 3.

R2: Figure 6 - It is not clear what data is being presented here. Are these only taxa present on both seamounts and nodules? Please clarify in the figure caption

A: Yes, indeed. We clarified the caption to: "Morphospecies present in both seamounts and nodule field transect and their average density (ind/100m) in each ecosystem."

Please also note the supplement to this comment:
https://www.biogeosciences-discuss.net/bg-2019-304/bg-2019-304-AC2-supplement.pdf

———————————————————

[Figure]

Average densities (ind/100m)

Pooled taxa

**Fig. 1.**

---

## Author Response (AR1)

January 2020

Subject: Revision of MS No.: bg-2019-304

Dear Associate editor,

We would like to thank both reviewers for their thorough revision of the manuscript.

We addressed the main concerns regarding Figure 3 and Table A1 and elaborated on certain parts of the methodology. Figures were redone to make sure the same dataset was used and the text was corrected whenever necessary. We argument the use of the ind/100m metric due to the limitation of the data collection, which is explained in more detail below.

All our other detailed replies to the reviewers' and associate editor's queries, the changes made to the manuscript (with line references of the revised manuscript) and a marked-up manuscript version are enclosed,

Sincerely,

Daphne Cuvelier

**Associate Editor Comments to the Author:**
Dear authors, reviewer comments have been received for your manuscript that you have both responded to. Both reviewers have made substantial comments that need to be addressed and incorporated during the revision of your manuscript. This should be possible considering your replies.

A few additional comments (in addition to the reviewer's comments) from my side are:
I agree with the reviewers that Figure S1 presents essential information on the size of seamounts, water depths and locations of the transects. Therefore it should be moved into the article. However, important information is not accessible from the current figure, because axes and color code labels are not readable (even if zooming into the PDF) due to too low resolution of the figure.

A: Figures are embedded in a pdf as requested when submitting a manuscript for Biogeosciences, which is why their resolution is not that high. We have increased the letter type so that the numbers are readable and increased the scales so that the size of the seamounts can be deduced. We will provide high resolution figures when the manuscript is accepted.

During the revision, please take care to significantly improve the quality of all figures (Fig.7 has a similar problem) that labels and annotations become readable.

A: See comment above. We reorganised figure7 to enhance its readability whilst embedded in the pdf.

It may be useful to add a table in the supplement containing specific information for each seamount raised by R#1 (i.e. size, summit and base depth, average slopes at sampling/video depths and average current direction/velocity (e.g. sampling site upstream or downstream), etc).
A: All this information can be found in the text and in the improved Fig 1. Slope can be calculated using the depth gradient and the length of the transect.

Sampling sites were all downstream – this is mentioned in L83-84 of the revised ms.

R#1, comment 3: The distance between the ROV laser spots on the videos (I believe it is 50 cm for the parallel beams, but please confirm with the ROV team of Kiel6000) should enable estimating the width of the visual field and hence allow providing an area for the transects, i.e. ind/m2. This would increase comparability.

A: This was considered previously and extensively, but lasers were not always visible during the entirety of the seamount dives. Changing camera pan and tilt and forward-looking view also introduce bias in any possible surface calculations (e.g. perspective view). Additionally, the non-constant travel speed, and changing ROV altitude complicates the surface calculations. In order to counter this, we considered subsampling images at a predefined distance interval (e.g. every 10 m) and extrapolate the surface covered. Nevertheless, as stated before, lasers were not visible/operational during the entire dive and would thus only increase over-or underestimation of the surface covered and therefore of the faunal densities. Overall, the ind/100m appears a more correct representation of the patterns observed and therefore the *in situ* reality and allowed for an in-detail comparison to the nodule field dives, which was one of the main objectives of this study.

R#1, comment L.288-295: This is an important point and I agree with the reviewer that more discussion on possible reasons would be helpful. For example, the seafloor POC flux in APEI3 seems to be reduced by ~30% compared to the investigated license areas (see biogeochemistry paper of Volz et al. (2018) Deep-Sea Research I 140).

A: We mitigated the overall statement that the nodule coverage would be the main driving factor and added more on possible other factors at play using the reference proposed. L329-331.

R#2, comment 3: Where is Fig. R1 you mention in your reply?
A: It was attached as a supplementary figure during the first review round and can be found in the "reply to reviewer 2" .pdf. We have inserted the figure in the appendix (Fig. A1) and referred to it in Section 3.2.

**Reviewer 1:**

**R1:** My main concern is that no environmental data are presented. In additon to missing hydrographical data, there is no detailed description of the sampling sites, particularly the seamounts, such as size, summit and base depth, inclination of slopes/general bathymetry, or current field. More important, no information is given on habitat types encountered along the transects. It is well known that substrate can vary considerably at short distances at seamounts, and of course megafaunal communities are strongly associated with substrate type. This is briefly mentioned in the discussion, but I do not understand why this information is not provided and analysed in the results. It should easily be available from the video footage.

**A:** Regarding the reviewer's comment on the description of the sampling sites: There was a supplementary figure (Fig. S1) from which the size of the seamount, the depth of the base and bathymetry can be deduced and which we now have incorporated into Figure 1 (as requested by the other reviewer and associate editor).

We have added information on the amount of hard substrata under the form of 3 categories: (1) Predominant soft substrata (<40% hard substrata), (2) mix or transition (between 40 and 60% hard substrata) and (3) predominant hard substrata (>60% hard substrata), annotated per 10m distance unit. This has been added to the methods section (L121-124). The amount of hard substrata was linked with faunal observations in higher taxonomic groups. No significant correlations between substratum type and faunal abundances were found for ROV02, ROV04 and ROV09. This is most likely due to the amount of hard/mix/soft substrata, which were almost equally distributed over the transects (varying between 16-39%, 30-51% and 15-38%, respectively). This highlights the need for a more in-detail assessment of substratum type, which is currently underway as the geomorphology of the seamount transects is being modelled (based on backscatter data) and investigated in detail, but this falls outside the scope of the current manuscript.

Based on the preliminary substratum results, a paragraph was added in section 3.1 L206-218

Concerning the hydrographical properties, a link with the POC-flux as known for the CCZ area was elaborated, also following the associate reviewer's suggestion (L328-332). Following the comments of the other reviewer, we have added the predominant current direction at the CCZ in the Methods section.

**R1:** There are also some methodological issues. The basic problem, as also admitted by the authors in the discussion, is that only a very limited number of rather short transects without replications are available and that transects at the seamounts and at the nodule fields were taken at different depths; in the case of Mann Borgese Smt the depth sampled was nearly 3000 m less than on the corresponding nodule field, and hence the data are hardly comparable. Although the depth difference at the other sites was much smaller, it may also limit the comparability of the data. This is mentioned in the discussion, but the consequences should be elaborated in more detail, and it makes the conclusion that "seamounts appear inadequate as refuge areas to help maintain nodule biodiversity" disputable.

**A:** In name of all the co-authors, I think we were very cautious in our conclusions, recognising the sampling shortcomings as well as the limited amount of data. We purposefully stated "*Based on our current knowledge*; seamounts appear inadequate as refuge areas to help maintain nodule biodiversity." And then urged on for more proper sampling to adequately corroborate or refute observations done here. We recognised the shortcomings throughout the manuscript. We think it is important to take a look at the entirety of the sentences written and to not take parts of them out of context.

Moreover, because of the unknown impacts and extent of mining, we could speculate that only the communities living outside a certain range in distance across the seafloor and upwards in the water column, will be protected from the mining plumes (and other impacts). Hence, it is relevant to also investigate "shallower" areas of the seamounts, as there is a possibility that recolonisation will start from these somewhat shallower areas.

**R1:** I am also not convinced that the quantification of the samples is correct. In section 2.1, the authors state that the altitude of the ROV was "kept constant whenever possible". Apart from not providing the information at which target altitude the ROV was kept, and whether this was the same at all transects, the authors inform in section 2.2 that, due to varying altitude as well as pan and tilt of the camera, "surface coverage" could not be used for standardisation and instead just transit length was used. However, since the field of view and thus the number of visible objects per unit transit section depend on the altitude and angle of the camera, the standardisation to 100 m transit sections, without taking into account the varying field of view, could strongly bias the results. Generally, the methods section has to be improved with much more detail.

**A:** Target altitude was 2m above seafloor and travel speed ~0.2m/s, though interrupted by sampling actions, instrument check-ups, exploration, object avoidance (in the case of the uphill seamount transects) etc. L100-102

Regarding the comment that ind/100m might not be a good metric, we would like to clarify that the length of the transects was calculated only taking into account the parts of the dive when the ROV was visualising the seafloor. The parts of the dives where the ROV was higher up in the water column (i.e. >10m altitude) and/or not visualising the seafloor (e.g. Transiting or checking ROV parts or instruments) whilst travelling forward, were omitted out of these calculations, thus granting a best estimative possible and allowing for comparison.

We agree with the reviewer's observation that changing altitude and angle of the camera bias the observations, but by excluding the parts of the transect that were too high up, it is the best approximation possible with the data at hand.

Throughout the paper, we acknowledge the importance of performing standardising video transects and recognise the shortcomings of our study to this respect. However, on seamounts we cannot fly straight all the time, it does not necessarily prevent comparisons between transects, because the bias is systematic across transects. Moreover, while this shortcoming may pose limitations to quantitative comparisons, it does not preclude qualitative comparison between habitats, which are the main objective of this study.

Information on ROV altitude and transects length calculations were added in the Methods section with the following paragraph:

*"For the transect length calculation for each dive, we omitted all parts of the video footage in which the ROV was at an altitude of >10m, or sections where the ROV was not visualising the seafloor (e.g. during transiting or inspecting ROV parts or instruments). Visualisation of ancient disturbance tracks were omitted as well, as these fell out of the scope of the article."* L128-132

**R1:** Further, it is not clear how the investigations made in this study relate to those by Vanreusel et al (2016) who also presented results for epifaunal communities in the CCZ, comparing APEI, BGR, GSR and others. Obviously the sampling was done on the same cruise using the same gear. Were the same nodule field transects analysed? If yes, this has to be justified, the additional value of this study as compared to Vanreusel et al (2016) has to be demonstrated (apart from the additional seamount transects) and any overlap and differences in the analysis indicated. If not, a thorough comparison between the results of both studies is necessary.

**A:** Vanreusel et al. was based on a subset of the BGR, GSR and APEI3 nodule field videotransects/dives analysed here (they investigated 2740 m of the 6100m nodule fields transects as presented here or 44% of our study**)**, and, as stated by the reviewer, they did not study the seamounts. Moreover, and most importantly, Vanreusel et al. did not identify individuals to morphospecies level but stayed at a higher taxonomical level such as Actiniaria, Echinoidea etc.

The following paragraph has been added in the ms for clarification purposes**,** in methods 2.2. L115-120

*"A subset of the nodule field transects form BGR, GSR and APEI3 was presented by Vanreusel et al. (2016), and correspond to 44% of what we studied here and limited organism identification to a higher taxonomic level (Order (e.g. Alcyonacea) or Class (e.g. Ophiuroidea)). In our study, the entire transects (100%) were annotated to morphospecies level, allowing a more detailed comparison between seamounts and nodule fields."*

Specific comments

**R1:** Use consequently "Ophiuroida, Asteroida" etc. instead of "ophiuroids, asteroids" etc.

**A:** Ok

**R1:** Abundances given in the text are not always consistent with those presented in Tab. A1. I did not check all entries, but two examples caught my eye: A total abundance of 89.2 ind/100 m is given for ROV10 in Tab. 1 and in the text, but summing up all observations in Tab. A1 results in ca. 67 ind/100 m. Another example: For Porifera, numbers given in the text match those in Tab. A1 for Rüppel and Senckenberg, but those presented for Heip and Mann Borgese are much lower than in the table (3 vs. 7.5 and 0.68 vs. 1.9, respectively). This has to be checked and resolved.

**A:** We carefully and meticulously checked all the densities, abundances and number of observations, to make sure they were correct and verified that the correct dataset was used for all figures and graphs.

**R1:** Line 42: Insert common definition for "seamount" and citation

**A:** We added the definition from the glossary of the International Seabed Authority: *"Seamounts are defined as isolated sub-surface topographic feature, usually of volcanic origin, of significant height above the seafloor (International Seabed Authority (ISA), 2019)"* on L42-43

https://www.isa.org.jm/scientific-glossary/

**R1:** 73: When did the sampling take place?

**A:** In 2015, this was added on L74

**R1:** 96: What is the difference between "exploration and opportunistic sampling"? More detail is needed.

**A:** The words chosen are rather self-explanatory. Explorative dives are dives when a site is visited for the first time and observations made during the dive are key to decide what happens during its course, e.g. sampling when the occasion presents itself or just imagery sampling. It was also mentioned in the text of the first submitted ms (L113-114) and now in L125-126.

**R1:** 98: What does "whenever possible" mean? 90 % of the transects? What was the target altitude of the ROV, and was it the same at all transects? How did panning and tilting affect the field of view? (see also general comments).

**A:** See answer above

**R1:** 94-99: Generally, much more information on the sampling mode is necessary, including sampling strategy (e.g., straight line, deviations for interesting objects etc.), ROV speed, target altitude, field of view etc.

**A:** See answer above

**R1:** 104: What is "ID's"? I guess it should be IDs, but "ID" is not defined in the text. Same in line 111.

**A:** ID from identification. This was altered.

**R1:** 114: Is there a reason that specimens collected were obviously not used for proper identification?

**A:** Samples were used for proper identification whenever possible. Multidisciplinary research cruises such as SO239 based on larger research projects (JPIO) tend to have a multitude of institutes involved, with different or overlapping interests. Samples taken during the cruise were distributed and divided over different institutes, each working towards their own objectives. The organisms sampled for which we received identifications were incorporated as such, e.g. Porifera identifications as included here. Though as stated in ms L280-282, even when organisms were sampled and identified, they were hard to extrapolate across the video imagery. Same reasoning applied for the Ophiuroidea where many species were revealed based on the samples, though impossible to annotate or differentiate based on the imagery footage (Christodoulou et al 2019). Hence, no information on its abundance, distribution or even presence in other areas than the one sampled can be included. There is no use of having a name for one (sampled) species from one single location if you cannot recognise it elsewhere.

**R1:** 118: Which statistical testing? Did the authors use tests other than nMDS? If yes, they have to be described here in detail

**A:** We have added information on the Kendall species Associations test carried out. L136-138

**R1:** 127: Here and throughout the text: two significant digits are sufficient, for example 7.6 instead of 7.59 or 89 vs. 89.23. The two decimals pretend a non-existing precision of the data.

**A:** Ok

**R1:** 151/152: Aren't Acrocirridae polychaetes as well? (". . . Acrocirridae were observed. . . as well."). Do you probably mean they were observed in high densities in some of the transects?

**A:** This was corrected.

**R1:** 189/190: This belongs into the discussion.

**A:** This is also mentioned in the discussion and it was mentioned here as well to recognise the limited sampling. This links back to the methodological issues as stated by the reviewer, which we fully recognise throughout the manuscript.

**R1:** 193: Insert ", respectively" after "Table A1"

**A:** Ok

**R1:** 195: "less linear" - how was this assessed? I cannot see any linear or non-linear relations in Fig. 4d, nor can I see any curves crossing.

**A:** Replaced linear by straightforward. Curves cross at smaller sample sizes (<100 individuals) for ROV13,08 and 10.

**R1:** 198: What does "small sample size" mean? I think the sample size in this study was always small.

**A:** Less than 100 individuals, this was added.

**R1:** 205: Should read "least overlap". Explain similarity between these findings and the results from the seamount: For both habitat types, the samples at APEI3 had least overlap with the other sites.

**A:** Ok

**R1:** 220ff: According to Fig. 3 (not Fig. 8!), the majority of ophiuroids on the nodule fields were unidentified

**A:** This was corrected to "*The majority of the very abundant Ophiuroidea observed at the CCZ seamounts were small and situated on hard substrata (morphospecies 5), while most of the Ophiuroidea at nodule fields (including morphospecies 6) were observed on the soft sediments. Morphospecies 6 was only rarely observed on the seamounts (Fig. 3)*" L253-256

**R1:** 244/245: This is not clear. Variation "along the video transects" was obviously not analysed and cannot be seen in Fig. 5. Probably the authors mean "between transects"?

**A:** This was corrected to "among the video transects of both seamounts and nodule fields"

**R1:** 251: Kendall's coefficient is not mentioned in methods section. See comment above.

**A:** This was added, see reply above

**R1:** 255-258: This makes no sense. If sampling depth differs between seamounts and NF, and nMDS distinguishes between seamount and NF groups, then the grouping must correspond also to depth sampled. Omit this paragraph (and Fig. 7b) and state in the discussion that differences between seamounts and NF could be a result of different depths sampled.

**A:** It was already stated in the discussion, but in our opinion it is a visual presentation of this statement, which is why we decided to keep it for now.

**R1:** 262: This is not quite clear. Rephrase: ". . . at different locations and additionally, for the seamounts, different depth ranges." Possible differences in substrate etc. should be mentioned here.

**A:** Change was carried out

**R1:** 269: Rephrase: ". . . since (mega)faunal communities could be very different even between adjacent seamounts . . .

**A:** Ok

**R1:** 270: Which parameters? Name examples for depth-dependent parameters which drive faunal composition

**A:** "parameters that vary with depth, such as temperature, oxygen concentration, substratum type, food availability, and pressure" This was added. L302-304

**R1:** 278: Why would "elevated topography (peaks)" favour Porifera and Anthipataria? Name possible mechanism(s). By the way: Seamounts are per definitionem elevated topography

**A:** Peaks are more exposed and appear thus more advantageous for filter feeders such as Porifera and Antipatharia. This is mentioned in the text. L311

**R1:** 288-295: Do the authors mean that faunal density is negatively correlated with nodule coverage? This is in contrast to Vanreusel et al 2016, who found higher abundances at higher nodule coverage. So obviously in this study, the driver for the differences in faunal density was not nodule coverage, but probably organic input.

**A:** We searched for a possible explanation as to why APEI3 stood out and found that this difference, besides their more northward location under more oligotrophic waters (mentioned in L328-332), corresponded to a difference in nodule coverage. The patterns by which nodule coverage or densities influence the ecological patterns are still poorly understood. The nodule coverage data as mentioned in our ms originate from Table S1-1 from Vanreusel et al. 2016, and indeed in the body of text these authors reported higher epifaunal densities in areas with dense nodule coverage, reporting >25 versus ≤10 in sessile individuals per 100 m2 for nodule rich and nodule free areas respectively. Nevertheless, if we compare the nodule coverage from Table S1 to figure 3 (both from Vanreusel et al. 2016) same patterns as those described in our study are observed, namely: Higher nodule coverage in APEI3 and lower densities both for sessile and mobile fauna. It is possible that Vanreusel et al. made their statement by looking at the license areas only and not included the APEI in this comparison. We chose to keep our statement, but elaborated on other possible reasons that could influence (e.g. POC) the patterns as observed (L328-331).

**R1:** 296: Grammar: neither - nor

**A:** Ok

**R1:** 319: Clearly distinguish between own data and data from literature by rephrasing, e.g. "Vanreusel et al. (2016) found that ophiuroids. . .."

**A:** Ok

**R1:** 322 : The available data cannot show a gradient, therefore it should read: ">50% less at seamounts compared to nodule fields"

**A:** This was corrected

**R1:** 331: ". . . studied here."

**A:** Ok

**R1:** 338: How can an uneven distribution (of holothuroids) affect composition?

**A:** Unevenly distributed organisms can give different perceptions in sampling. Organisms with a wide distribution range can, when unevenly distributed, be present/absent in adjacent sampling localities, thus resulting in different faunal composition for these sampling locality.

**R1:** 350: This is an isolated statement here - what does it imply?

**A:** We moved it to the beginning of the paragraph which now starts off as follows: "Stalked organisms, such as Crinoidea (Echinodermata) and Hexactinellida (except for Amphidiscophora, Porifera) rely on hard substrata for their attachment and are considered being among the most vulnerable organisms when mining is concerned." L381-383

**R1:** 352: And what about nodule-covered areas - did they host these taxa in Vanreusel's or this study? This paragraph is a bit confusing and should be re-sorted, also clearly distinguishing between own results and those of others.

**A:** This paragraph was re-written (taking into account the comments of the other reviewer as well) to make the distinction between our results and those from literature more easily: L391-402

**R1:** 361: "communities"

**A:** Ok

**R1:** 362: ". . . were more abundant. . ." - compared to what?

**A:** "… more abundant than on seamounts". This was added.

**R1:** 370: ". . .they are known to . . ." Citation?

**A:** This statement was based on our personal observations. We have rephrased it and changed the structure of the paragraph as to convey our point more clearly:  L411-417

**R1:** 372: Does the reference (Baco 2007) apply to both statements? I suggest to rephrase, e.g. "The exception. . .common on seamounts, as also reported in other studies (e.g., Baco. 2007)." Baco 2007 is not in the reference list!

**A:** Baco 2007 refers to the Scleractinia being common on seamounts. We have clarified this and added the reference for Baco 2007 to the reference list. L415-417

**R1:** 373: Insert: ". . . Enteropneusta which in this study were found only on seamounts, were. . ."

**A:** Ok

**Figures**

**R1:** Fig. 1: What does the "A" in the upper left corner mean?

**A:** Figure 1 underwent some change taking into account the comments from the other reviewer.

**R1:** Fig. 2: I suggest to add morphospecies "names" (as given in Fig. 3) to the examples.

**A:** Morphospecies names were added to the caption.

**R1:** Fig. 3: This figure should be simplified. Most of the morphospecies were observed in very low numbers, and in these cases differences between NF and seamounts are difficult to see in the figure and rather not relevant. I suggest to include in this figure only morphospecies and higher taxa with a substantial mean abundance (e.g., >0.5 ind./100 m per habitat type); other morphospecies could be summarised or omitted. By contrast, Table A1 should be extended and present the results for all morphospecies, not only higher taxa (see below).

**A:** Figure 3 was withheld because it was considered a key figure to show how different the presence/absence/abundance of the fauna varies between the two ecosystems, but the 3 parts were separated more clearly with different breaks at the X-axis, thus enhancing readability and interpretability. We also reorganised figure 3 to correspond to the order of A Table 1.

Table A1 now includes the morphospecies densities as well.

**R1:** Fig. 4: Axis labels are incomplete (units are missing). What does "exact" on the y-axis in panels a and c mean? And what is sample size (units?) in panels b and d? I guess that not sample size was used for the rarefaction curves, but accumulated number of observations. Caption is incomplete: What do the shaded areas in panels a and c and horizontal and vertical lines in panels b and D mean?

**A:** Sample size is the number of individuals observed (or number of observations as you will). Information was added to the caption and axis were renamed and/or clarified. Horizontal lines of the lower panels were omitted because they did not provide significant information for the interpretation of the figure.

**R1:** Fig. 5: What does "values are relative" mean? - percent (of what?)? This has to be explained.

**A:** Values are relative due to different transect lengths and differences in richness. This was changed to "Values are indicative rather than absolute due to different transect lengths and differences in richness."

**R1:** Fig. 6: y-axis incomplete, should include quantity and unit.

**A:** I am not sure what this is about, since the Y-axis is complete and has ind/100m as title.

**R1:** Fig. 7: Omit panel b).

**A:** See comment above

**Tables**

**R1:** Tab. 1: Be consistent with units: here, #obs/100 m is given, whereas throughout the text and in figures and in Tab. A1 the unit for density is ind/100 m.

**A:** The number (#) of observations is more of a methodological way to describe it, since it was used prior to the results. This could be changed.

**R1:** See also comment to Fig. 3. I suggest to list data for all distinguished morphospecies here and sums for higher taxa. It is irritating that densities for higher taxa (e.g., Holothuroidea) are given, but they do not include the identified morphospecies within that taxon. Not identified taxa should be clearly indicated, e.g. Holothuroidea indet., and they should sum up with the distinguished morphospecies to total Holothuroidea, etc. I also suggest to include absolute number of observations in addition to densities in this table. This would facilitate the evaluation whether, e.g., differences are based on a substantial number of individuals, or rely on just one or two specimens. The order of taxa in the table is not clear

**A:** Morphospecies densities (ind/100m) have been added to the table. We opted for densities to correspond to the data used in the manuscript. Absolute numbers are easy to deduce since the length of the transects are given in Table 1. The order of taxa in the table was reorganised

**Reviewer 2:**

**Major comments**

**R2:** 1) The appendix fig1 is a very important figure to place all the observations into context. I would move it from an appendix to a regular figure for the paper or as new panels in existing Fig. 1.

**A:** We have incorporated it as an extra panel in Figure 1.

**R2:** 2) The data presentation used to compare nodule transects to seamount transects should be refined. Right now figure 3 portrays averages of densities for fine taxa which, based on the finding of very low overlap between transects, means little - averaging a large number for one transect with low or zero numbers in other transects. Rather the data should be presented at broader taxonomic categories (as in Appendix table 1) with average (and standard deviation) densities and # of morphospecies. This would also follow the language in the results section better. Data on each fine morphospecies (level of taxonomy in Fig3) could be presented an an appendix and by transect.

**A:** We decided to add morphospecies to Table A1 (also taking into account the comments made by the other reviewer), which thus give information per transect, help to elucidate Fig. 3 and add the desired level of taxonomy. The order of morphospecies along the y-axis of figure 3 was also reorganised to correspond to that of Table A1. Figure 3 was thus withheld because it was considered a key figure to show how different the presence/absence/abundance of the fauna varies between the two ecosystems, but the 3 parts were separated more clearly and different breaks at the X-axis, thus enhancing readability and interpretability. This information was added in the figure caption as well.

**R2:** 3) Once data is pooled at higher taxa levels, statistical comparisons could be drawn to compare the average # morphospecies and average density between nodule and seamount transects.

**A:** There were no significant differences for the densities (T-tests for samples with unequal variance) per taxon (taxa pooled and tested: Actiniaria, Alcyonacea, Ceriantharia, Corallimorpharia, Antipatharia, Pennatulacea, Scleractinia, Zoantharia, Bryozoa, Asteroidea, Crinoidea, Echinoidea, Holothuroidea, Ophiuroidea, Porifera, Tunicata, Enteropneusta, Gastropoda. The pooled data were visualised as a histogram with st. dev. and added as an extra figure to this reply to the reviewer. We added the figure to the manuscript as Fig. A1 in the appendix and referred to it in Section 3.2.

The number of morphospecies per higher pooled taxon group proved more difficult, since often we know there are >1 or >2 morphospecies, but not the exact numbers. Using the minimum known number would be and under-estimation, which is why we chose not to test these pooled observations.

This comment was also taking into account for the overlapping morphospecies in Figure 6, for which standard deviation bars were added.

**R2:** 4) The authors conclude that the ratio of hard/soft habitat may explain some of the faunal trends they observe. Can't this ratio be determined from the transects? If possible add this metric to help explain faunal communities along or between the transects.

**A:** We have added information on the amount of hard substrata under the form of 3 categories: (1) Predominant soft substrata (<40% hard substrata), (2) mix or transition (40-60% hard substrata) and (3) predominant hard substrata (>60% hard substrata), annotated over 10m distance units. Very few significant relationships were revealed (only for Mann Borgese ROV15), though backscatter data is currently being analysed to model the geomorphology along the transects in more detail and help reveal more details of fauna/substratum relationships, but it is out of the scope of this article.

The following paragraph was added in section 3.1 (also taking into account the other reviewer's suggestions): L205-2017

*"About 57% of all sessile fauna was associated with predominantly hard substrata, followed by 31% on the mixed substrata. For the mobile taxa the pattern was less pronounced with 41 and 42% associated with predominantly hard and mixed soft/hard substrata respectively. The amount of predominantly hard and soft substrata were negatively correlated though, not significantly. This was due to the elevated amount of mixed hard/soft substrata featuring equal amounts 40-60%. Over all seamount transects pooled together, no taxa were significantly correlated with the amount of hard substrata, nor with soft substrata. When looking at the individual transects, no significant correlations were found between taxa and substrata for ROV02 or ROV04 or ROV09, most likely due to the equal distribution of the amount of hard/soft/mix substrata. In this perspective, ROV15 stood out, as it was dominated by predominantly hard substrata (56/%): For this transect, Pennatulacea were significantly negatively correlated with the amount of hard substrata and Zoantharia/Octocorralia were significantly and positively correlated with hard substrata, as were Ophiuroidea, Asteroidea, Crinoidea and Mollusca."*

**Specific comments**

**R2:** line 82 - Explain why the north or northwestern flanks of the seamounts were chosen for the transects.

**A:** These flanks were chosen based on the positioning of the vessel and the predominant surface current in order to avoid the umbilical of the ROV to drift/being transported towards the vessel. Predominant currents in the CCZ are South-East oriented, allowing for an ROV positioning "downstream" of the vessel's location while visiting the north-northwestern seamount flanks. This was added in L81-84

**R2:** line - 97-99 - Provide the range of altitude, speed that were kept constant.

**A:** Target altitude was <2m above seafloor and travel speed ~0.2m/s, though interrupted by sampling actions, instrument check-ups, exploration, object avoidance (in the case of the uphill seamount transects) etc. This was added in the body of text. L99-102

**R2:** line 208-209 - this statement appears true when examining the higher taxa pooled data in appendix table 1. however in the cited fig 3, its hard to actually make this comparison because averages at finer taxonomic categories are highly variable due to lack of fine taxa overlap between transects.

**A:** This is amended by adding the morphospecies information in Table A1 and reorganising figure 3 to correspond to Table A1.

**R2:** line 267 - Start the sentence with, "Amongst the seamount transects,..."

**A:** Ok

**R2:** line 269 - The point of this sentence is not clear as it opposes the trend you find. Clarify.

**A:** This sentence was to point out that it could rather be depth influencing their similarity than their adjacent location. This part was changed (also taking into account the comments from the other reviewer) L298-303

**R2:** lines 296-306 - Nice to see a paragraph which lays out what future transecting should look like. The paragraph mentioned that wider depth ranges should be included and the data and literature certainly support that. Might it also be wise to have transects that move along countours so there are many replicate obsrevations at a given depth, instead of conducting uphill transects? Adding a sentence or two addressing across slope vs with slope transecting would be worthwhile.

**A:** This is a valid observation which was added in L340-342

**R2:** line 308 - The sentence should be slightly reworded based on the authros findings to "Seamounts were shown to share FEW fauna with surrounding habitats...."

**A:** Based on the literature, fauna from seamounts tend to occur in neighbouring habitats quite easily, but that is not the case here. We rephrased the sentence to clarify this. L345-347

**R2:** Line 316 - this topic sentence needs to be improved. Rather than simply reiterating the results section can this paragraph be rewritten and a topic sentence created that summarizes the functional differences between taxa on seamounts vs nodules?. See literature by Rowden etal that look at functional variation of taxa on seamounts and neighboring areas. E.g. Rowden et al 2010 Marine Ecology

**A:** We added in topic sentences for several paragraphs and cleaned up the remainder of the body of text as to not repeat the results. L353-409

On a side note, functional traits of seamount and nodule field fauna are being investigated in a broader framework, extending beyond the feeding group and including life history, mobility etc.

**R2:** Table 1 - Add "SM:" before Mann Borgese

**A:** This was added to the table.

**R2:** figure 3 - given that there is so little overlap in the morphospecies between each sampled transect. Figure 3 is a bit hard to interpret. Error bars would help. Its great that the taxonomic diversity is presented but this might be better in the appendix.

**A:** We changed the Y-axis of the different parts of the figure 3 graph to make them more easily interpretable. See reply above. We have uploaded an extra figure linked to this reply to the reviewers that was included in the appendix, pooling the densities into larger taxa and referred to in section 3.2.

**R2:** Instead, appendix table 1 which presents higher taxa and # morphospecies might be the better data to show in the main paper. Here densities at higher levels can be better compared. Averageing the abundances across the transects seems ill advised given the differences observed between each one.

**A:** We have added the morphospecies' densities to the appendix table, which also clarifies the taxonomy and figure 3.

**R2:** Figure 6 - It is not clear what data is being presented here. Are these only taxa present on both seamounts and nodules? Please clarify in the figure caption

**A:** Yes, indeed. We clarified the caption.

[revised manuscript text omitted]

Ophiuroidea at nodule fields (abundant morphospecies at nodule fields (including morphospecies 6)
wereas mostly observed on the soft sediments of the nodule transects. This mMorphospecies 6 was
only rarely observed on the seamounts (Fig. 3). Another easily recognisable morphospecies was
found on Porifera, corals and animal stalks and was more abundant at seamounts than at nodule
fields (morphospecies 4) (Fig. 2 and 3).

Crinoidea, Asteroidea (both Echinodermata) and Antipatharia (Cnidaria) were more abundant on the
seamounts (Fig. A1). This coincided with a higher diversity for Asteroidea and Antipatharia on the
seamounts as well. Crinoidea diversity was similar (5 to 4 morphospecies comparing seamounts to
nodule fields). Holothuroidea occurred in similar densities in both ecosystems (Fig. A1, though they
were characterised by different morphospecies (Fig. 3). Overall densities of Echinoidea were highest
comparable between seamounts andat nodule fields, though for the nodule fields this this was
mostly due to one very abundant morphospecies, namely Aspidodiadematidae msp 1, which was
absent at the seamounts (Fig. 3). Besides this one very abundant morphospecies, which was only
present at nodule fields, Echinoideaechinoids showed higher densities at seamounts and were more
diverse at seamounts (11 morphospecies vs. 5 at nodule fields).

There was no morphospecies overlap for Tunicata, Antipatharia, and Actiniaria. Alcyonacea,
Ceriantharia, Corallimorphidae and Crinoidea only shared 1 morphospecies between seamounts and
nodule fields, namely *Callozostron* cf. *bayeri*, Ceriantharia msp. 2, *Corallimorphus* msp. 2 and
Comatulida msp. 1 respectively (Fig. 6).

There were no observations of Enteropneusta, Scleractinia and Zoantharia (Cnidaria), Aphroditidae
(Polychaeta) or holothuroid Deimatidae at the nodule fields transects (Table A1, Fig. A1). While
Actinopterygii were left out of the analysis, it should be noted that fish observations were more
abundant and diverse at the seamounts than on the nodule fields (Table A1).

There was quite some faunal variation observed almong the video transects carried out in the

[revised manuscript text omitted]

Font: 11 pt

| Page 57: [2] Formatted | Daphne Cuvelier | 16/12/2019 15:03:00 |
| --- | --- | --- |

Font: 11 pt

| Page 57: [3] Formatted | Daphne Cuvelier | 16/12/2019 15:03:00 |
| --- | --- | --- |

Font: 11 pt

| Page 57: [4] Formatted Table | Daphne Cuvelier | 16/12/2019 15:30:00 |
| --- | --- | --- |

Formatted Table

| Page 57: [5] Formatted | Daphne Cuvelier | 16/12/2019 15:03:00 |
| --- | --- | --- |

Font: 11 pt

| Page 57: [6] Formatted | Daphne Cuvelier | 16/12/2019 15:03:00 |
| --- | --- | --- |

Font: 11 pt

| Page 57: [7] Formatted | Daphne Cuvelier | 16/12/2019 15:03:00 |
| --- | --- | --- |

Font: 11 pt

| Page 57: [8] Formatted | Daphne Cuvelier | 16/12/2019 15:03:00 |
| --- | --- | --- |

Font: 11 pt

| Page 57: [9] Formatted | Daphne Cuvelier | 16/12/2019 15:03:00 |
| --- | --- | --- |

Font: 11 pt

| Page 57: [10] Formatted | Daphne Cuvelier | 16/12/2019 15:03:00 |
| --- | --- | --- |

Font: 11 pt

| Page 57: [11] Formatted | Daphne Cuvelier | 16/12/2019 15:03:00 |
| --- | --- | --- |

Font: 11 pt

| Page 57: [12] Formatted | Daphne Cuvelier | 16/12/2019 15:03:00 |
| --- | --- | --- |

Font: 11 pt

| Page 57: [13] Formatted | Daphne Cuvelier | 16/12/2019 15:03:00 |
| --- | --- | --- |

Font: 11 pt

| Page 57: [14] Formatted Table | Daphne Cuvelier | 16/12/2019 15:30:00 |
| --- | --- | --- |

Formatted Table

| Page 57: [15] Formatted | Daphne Cuvelier | 16/12/2019 15:03:00 |
| --- | --- | --- |

Font: 11 pt

| Page 57: [16] Formatted | Daphne Cuvelier | 16/12/2019 15:03:00 |
| --- | --- | --- |

Font: 11 pt

| Page 57: [17] Formatted | Daphne Cuvelier | 16/12/2019 15:03:00 |
|---|---|---|

Font: 11 pt

| Page 57: [18] Formatted Table | Daphne Cuvelier | 16/12/2019 15:30:00 |
|---|---|---|

Formatted Table

| Page 57: [19] Formatted | Daphne Cuvelier | 16/12/2019 15:03:00 |
|---|---|---|

Font: 11 pt

| Page 57: [20] Formatted | Daphne Cuvelier | 16/12/2019 15:03:00 |
|---|---|---|

Font: 11 pt

| Page 57: [21] Formatted | Daphne Cuvelier | 16/12/2019 15:03:00 |
|---|---|---|

Font: 11 pt

| Page 57: [22] Formatted | Daphne Cuvelier | 16/12/2019 15:03:00 |
|---|---|---|

Font: 11 pt

| Page 57: [23] Formatted Table | Daphne Cuvelier | 16/12/2019 15:30:00 |
|---|---|---|

Formatted Table

| Page 57: [24] Formatted | Daphne Cuvelier | 16/12/2019 15:03:00 |
|---|---|---|

Font: 11 pt

| Page 57: [25] Formatted | Daphne Cuvelier | 16/12/2019 15:03:00 |
|---|---|---|

Font: 11 pt

| Page 57: [26] Formatted | Daphne Cuvelier | 16/12/2019 15:03:00 |
|---|---|---|

Font: 11 pt

| Page 57: [27] Formatted | Daphne Cuvelier | 16/12/2019 15:03:00 |
|---|---|---|

Font: 11 pt

| Page 57: [28] Formatted | Daphne Cuvelier | 16/12/2019 15:03:00 |
|---|---|---|

Font: 11 pt

| Page 57: [29] Formatted | Daphne Cuvelier | 16/12/2019 15:03:00 |
|---|---|---|

Font: 11 pt

| Page 57: [30] Formatted | Daphne Cuvelier | 16/12/2019 15:03:00 |
|---|---|---|

Font: 11 pt

| Page 57: [31] Formatted | Daphne Cuvelier | 16/12/2019 15:03:00 |
|---|---|---|

Font: 11 pt

| Page 57: [32] Formatted | Daphne Cuvelier | 16/12/2019 15:03:00 |
|---|---|---|

Font: 11 pt

| Page 57: [33] Formatted Table | Daphne Cuvelier | 16/12/2019 15:30:00 |
|---|---|---|

Formatted Table

| Page 59: [34] Formatted | Daphne Cuvelier | 16/12/2019 15:03:00 |
|---|---|---|

Font: 11 pt

| Page 59: [35] Formatted | Daphne Cuvelier | 16/12/2019 15:03:00 |
|---|---|---|

Font: 11 pt

| Page 59: [36] Formatted | Daphne Cuvelier | 16/12/2019 15:03:00 |
|---|---|---|

Font: 11 pt

| Page 59: [37] Formatted | Daphne Cuvelier | 16/12/2019 15:03:00 |
|---|---|---|

Font: 11 pt

| Page 59: [38] Formatted | Daphne Cuvelier | 16/12/2019 15:03:00 |
|---|---|---|

Font: 11 pt

| Page 59: [39] Formatted | Daphne Cuvelier | 16/12/2019 15:03:00 |
|---|---|---|

Font: 11 pt

| Page 59: [40] Formatted | Daphne Cuvelier | 16/12/2019 15:03:00 |
|---|---|---|

Font: 11 pt

| Page 59: [41] Formatted | Daphne Cuvelier | 16/12/2019 15:03:00 |
|---|---|---|

Font: 11 pt

| Page 59: [42] Formatted | Daphne Cuvelier | 16/12/2019 15:03:00 |
|---|---|---|

Font: 11 pt

| Page 59: [43] Formatted Table | Daphne Cuvelier | 16/12/2019 15:33:00 |
|---|---|---|

Formatted Table

| Page 59: [44] Formatted | Daphne Cuvelier | 16/12/2019 15:03:00 |
|---|---|---|

Font: 11 pt

| Page 59: [45] Formatted | Daphne Cuvelier | 16/12/2019 15:03:00 |
|---|---|---|

Font: 11 pt

| Page 59: [46] Formatted | Daphne Cuvelier | 16/12/2019 15:03:00 |
|---|---|---|

Font: 11 pt

| Page 59: [47] Formatted | Daphne Cuvelier | 16/12/2019 15:03:00 |
|---|---|---|

Font: 11 pt

| Page 59: [48] Formatted | Daphne Cuvelier | 16/12/2019 15:03:00 |
|---|---|---|

Font: 11 pt

| Page 59: [49] Formatted Table | Daphne Cuvelier | 16/12/2019 15:33:00 |
|---|---|---|

Formatted Table

| Page 59: [50] Formatted | Daphne Cuvelier | 16/12/2019 15:03:00 |
|---|---|---|

Font: 11 pt

| Page 59: [51] Formatted Table | Daphne Cuvelier | 16/12/2019 15:33:00 |
|---|---|---|

Formatted Table

| Page 59: [52] Formatted | Daphne Cuvelier | 16/12/2019 15:03:00 |
|---|---|---|

Font: 11 pt

| Page 59: [53] Formatted | Daphne Cuvelier | 16/12/2019 15:03:00 |
|---|---|---|

Font: 11 pt

| Page 59: [54] Formatted | Daphne Cuvelier | 16/12/2019 15:03:00 |
|---|---|---|

Font: 11 pt

| Page 59: [55] Formatted | Daphne Cuvelier | 16/12/2019 15:03:00 |
|---|---|---|

Font: 11 pt

| Page 59: [56] Formatted Table | Daphne Cuvelier | 16/12/2019 15:33:00 |
|---|---|---|

Formatted Table

| Page 59: [57] Formatted | Daphne Cuvelier | 16/12/2019 15:03:00 |
|---|---|---|

Font: 11 pt

| Page 59: [58] Formatted | Daphne Cuvelier | 16/12/2019 15:03:00 |
|---|---|---|

Font: 11 pt

| Page 59: [59] Formatted | Daphne Cuvelier | 16/12/2019 15:03:00 |
|---|---|---|

Font: 11 pt

| Page 59: [60] Formatted | Daphne Cuvelier | 16/12/2019 15:03:00 |
|---|---|---|

Font: 11 pt

| Page 59: [61] Formatted | Daphne Cuvelier | 16/12/2019 15:03:00 |
|---|---|---|

Font: 11 pt

| Page 59: [62] Formatted | Daphne Cuvelier | 16/12/2019 15:03:00 |
|---|---|---|

Font: 11 pt

| **Page 59: [63] Formatted Table** | **Daphne Cuvelier** | **16/12/2019 15:33:00** |

Formatted Table

| **Page 59: [64] Formatted** | **Daphne Cuvelier** | **16/12/2019 15:03:00** |

Font: 11 pt

| **Page 59: [65] Formatted** | **Daphne Cuvelier** | **16/12/2019 15:03:00** |

Font: 11 pt

| **Page 59: [66] Formatted** | **Daphne Cuvelier** | **16/12/2019 15:03:00** |

Font: 11 pt

---

## Author Response (AR2)

Horta, 9 March 2020

Subject: Revision of MS No.: bg-2019-304

Dear Associate editor,

We understand the recurrent issue with the transect lengths vs. the surface estimations, also for comparison purposes with other studies.

Since laser pointers were 6.5 cm apart, these were not visible when altitudes were higher than 2m. We decided to forego the attempts to quantify the actual or exact amount of surface covered and estimated the approximate area covered by using the ROV altitude, time spent at altitude and distance travelled along with predefined image widths (in concordance to Vanreusel et al. (2m at 1m altitude and 4m at 2m altitude, and extrapolated from there)). The methodology for this was added in L126-136. Main results did not change significantly (except for a significant difference between Porifera densities between seamounts and nodule fields L229-230) and main tendencies were withheld.

Figures 3, 6 and 7 and Table A1 were altered accordingly and previous comments regarding figure 3 by this reviewer were re-considered and tests re-done (see below). This has some implications in table and figure numeration which are explained in our detailed comments below.

Sincerely,

Daphne Cuvelier, on behalf of all co-authors

Evaluation of bg-2019-304-manuscript-version3

**Methods**
R1: Lines 96 ff: I agree that the method for quantification of the samples may be suitable for the comparisons made in this study, provided that the variations in altitude and camera angle were comparable between transects (but I guess that keeping the target altitude on the flat plain was much easier than at the seamounts?). My only concern is the statement that "altitudes >10 m were omitted", meaning that, the other way round, all altitudes between <2 m and 10 m were considered? Since the area correlates with the square of the distance, the area observed could vary by a factor of >25 between sections, which would introduce a substantial bias.
Further, although the standardisation to section transect length is perfectly suitable for the comparisons within the study, it would be helpful to include some information on the approximate field of view of the camera, e.g. at the target altitude with a standard pan and tilt setting, in order to get at least some feeling when comparing this study with others.

A: We understand the recurrent issue with the transect lengths vs. the surface estimations, also for comparison purposes, and decided to estimate the approximate surface covered, as explained above. The bias caused by not taking into account the camera's zoom and pan and tilt is recognised in L137-138

Figures 3, 6 and 7 were altered accordingly and previous comments regarding figure 3 by the reviewer were re-considered and tests re-done (see below).

R1: The authors should also include the information given in their rebuttal, that specimens samples were also used for identifications when possible.

A:This was added in L107-108

**Results, Discussion**
R1: Line 202: delete "#".

A: # was replaced by "n=". This was done for all other cases in the ms as well.

R1: Lines 252ff (original 220): the modified statement is still not correct. The majority of the ophiuroids were not morphospecies 5, but Ophiuroidea indet.

A: This sentence was rephrased to: "Three Ophiuroidea morphospecies were present at both seamounts and nodule fields (Fig. 2, 3 and 6). Most of the Ophiuroidea observed at the CCZ seamounts that could be identified to morphospecies level, were small and situated on hard substrata (morphospecies 5), while those at nodule fields (including morphospecies 6) were observed on the soft sediments."

R1: Line 333: should read "…showed that an asymptote was reached neither …."

A: Change carried out

R1: Line 339: should read "1000 m"

A: Ok

**Figures and tables**

R1: Fig. 3: I am still not quite happy with this figure, because rare taxa are hardly comparable, despite modifying the axes, and I still suggest to sum up the less abundant morphospecies.

Apart from this, the x-axis labels are incomplete; should read: "Density (ind/100 m)". Negative density values for the seamounts are strange; the minus sign should be deleted. In panel (B), the scaling of the x-axis is not clear; only "-12" is given for the left hand part. I suggest to include a finer scaling here, such as 0.5, 1.0, 1.5 etc. Similar in panel (c). Caption: what does #4 and #5 mean - number of video transects? Rather write "4 transects" or "N=4".
A: Taking these comments into account, we decided to include figure A1 in the ms instead of figure 3 which is now an appended figure. We have added the minimum number of morphospecies to the new figure 3 as well, to facilitate discussing the results. And Table A1 passes now to be Table 2 in the ms. We still believe the back-to-back histogram figure holds valuable information, which is why it is still included in the appendix, though we understand the reviewer's point regarding the rare morphospecies. Figure references have been changed accordingly.

Axes and captions were altered as requested by the reviewer

R1: Fig. 4: Y-axis labels. In panels a and c: "Number of species" (with capital N). And what does "Species" in panels b and c mean? I guess it should also be "Number of species"?
A: change was carried out

R1: Fig. 6: my previous comment is still valid, the y-axis label is incomplete and should comprise quantity and unit; in this case: "Density (ind/100 m)"

A: Change was carried out

R1: Tab. A1: include sums for higher taxa; e.g., Cnidaria, Anthozoa, Ceriantharia etc.

Because of the presence of the new figure 3 and the readability and interpretability of the table, we did not include the sums on higher taxa levels in its current lay-out. It would be confusing to use sums and densities across the different taxonomic levels in the table.

[revised manuscript text omitted]